# SIRT6 overexpression in the nucleus protects mouse retinal pigment epithelium from oxidative stress

Xue Yang*, Jin-Yong Chung*, Usha Rai, Noriko Esumi

Retinal pigment epithelium (RPE) is essential for the survival of retinal photoreceptors. To study retinal degeneration, sodium iodate (NaIO$_3$) has been used to cause oxidative stress-induced RPE death followed by photoreceptor degeneration. However, analyses of RPE damage itself are still limited. Here, we characterized NaIO$_3$-induced RPE damage, which was divided into three regions: periphery with normal-shaped RPE, transitional zone with elongated cells, and center with severely damaged or lost RPE. Elongated cells in the transitional zone exhibited molecular characteristics of epithelial–mesenchymal transition. Central RPE was more susceptible to stresses than peripheral RPE. Under stresses, SIRT6, an NAD$^+$-dependent protein deacylase, rapidly translocated from the nucleus to the cytoplasm and colocalized with stress granule factor G3BP1, leading to nuclear SIRT6 depletion. To overcome this SIRT6 depletion, SIRT6 overexpression was induced in the nucleus in transgenic mice, which protected RPE from NaIO$_3$ and partially preserved catalase expression. These results demonstrate topological differences of mouse RPE and warrant further exploring SIRT6 as a potential target for protecting RPE from oxidative stress-induced damage.

## Introduction

The retinal pigment epithelium (RPE) is a single layer of pigmented cells with a cobblestone-like appearance and resides between retinal photoreceptor cells and the choroid of the eye (Strauss, 2005). The RPE directly contacts with photoreceptor outer segments and plays critical roles that support the health and function of photoreceptors. Without healthy RPE, therefore, normal vision cannot be achieved (Strauss, 2005). RPE cells are postmitotic and terminally differentiated; however, in pathological conditions, they lose the epithelial integrity and dedifferentiate, leading to the loss of mature functions (Zhao et al, 2011; Tamiya & Kaplan, 2016; Shu et al, 2020; Zhou et al, 2020). The RPE constantly faces oxidative stress because of daily phagocytosis of photoreceptor outer segments and its localization next to the high blood flow of the choroid, resulting in accumulated oxidative damage with age, which is thought to contribute to developing age-related macular degeneration (AMD), the leading cause of blindness in the elderly (Wong et al, 2014; Datta et al, 2017; Fleckenstein et al, 2021).

A variety of stresses including oxidative stress and cell dissociation can induce dedifferentiation of RPE cells (Kim et al, 2008; Zhao et al, 2011; Yang et al, 2018; Shu et al, 2020). Dissociated cultured RPE cells or cells migrating out of cultured RPE sheets have been shown to dedifferentiate into fibroblast-like cells through epithelial–mesenchymal transition (EMT) (Tamiya et al, 2010; Sripathi et al, 2021a, 2021b). In EMT, epithelial cells lose cell–cell contacts with neighboring cells and become fibroblastic mobile cells with increased mesenchymal markers (Lamouille et al, 2014; Dongre & Weinberg, 2019). There is a wide range of EMT from partial to complete EMT (Dongre & Weinberg, 2019). RPE cells undergoing EMT contribute to scarring and wound contractions in proliferative vitreoretinopathy (PVR) (Tamiya & Kaplan, 2016). In addition, EMT can result from oxidative stress, autophagy defects, and mitochondrial dysfunction, all of which are suspected as the mechanisms of AMD pathogenesis (Kim et al, 2008; Zhao et al, 2011; Ghosh et al, 2018; Yang et al, 2018; Datta et al, 2023).

To study photoreceptor degeneration, NaIO$_3$, an oxidizing reagent that is primarily toxic to the RPE in vivo, has long been used to cause RPE damage followed by photoreceptor death (Kannan & Hinton, 2014; Zhang et al, 2021). Many previous studies used a high dose of NaIO$_3$ to destroy the RPE and thereby create photoreceptor degeneration (Grignolo et al, 1966; Enzmann et al, 2006; Yang et al, 2014; Chowers et al, 2017). Thus, detailed characterization of NaIO$_3$-induced RPE damage itself is still limited. Using RPE flat-mounts with phalloidin staining that outlines the RPE cell shape, Xia et al. reported that NaIO$_3$ injection in mice produced three regions with distinct RPE morphologies: normal cobblestone-like RPE (periphery), atrophic damaged RPE (center), and irregular elongated cells between the two regions (transitional zone) (Xia et al, 2011). Recently, several others and we also observed similar RPE morphologies induced by NaIO$_3$ in mice (Ma et al, 2020; Upadhyay et al, 2020; Wolk et al, 2020; Yang et al, 2021; Zhang et al, 2021).

Wilmer Eye Institute, Johns Hopkins University School of Medicine, Baltimore, MD, USA

Correspondence: nesumi1@jhmi.edu
*Xue Yang and Jin-Yong Chung contributed equally to this work

SIRT6 is a member of the mammalian sirtuin family (SIRT1–7) of NAD⁺-dependent protein deacetylases (Tasselli et al, 2017; Chang et al, 2020). SIRT6 is multifunctional and acts as an NAD⁺-dependent protein deacetylase, deacylase, and mono-ADP ribosyltransferase in a variety of biological contexts, including inflammation, DNA repair, metabolism, oxidative stress, aging, and longevity (Mostoslavsky et al, 2006; Chang et al, 2020). However, there have been few studies of SIRT6 in the RPE, except recent reports related to autophagy (Feng et al, 2018; Liu & Liu, 2020). Among SIRT6's functions, the protective effect against oxidative stress could arise from (i) suppression of nuclear factor kappa B (NF-κB) activity, (ii) up-regulation of antioxidant genes through nuclear factor erythroid 2 like 2 (NFE2L2, also known as NRF2), and (iii) stimulation of DNA repair. For (i), SIRT6 binds to the RELA (p65) subunit of NF-κB and thereby is recruited to RELA target promoters, resulting in the suppression of NF-κB target genes by deacetylation of histone H3 lysine 9 (H3K9) (Kawahara et al, 2009, 2011). NF-κB, a master regulator of inflammation and immune responses, is also one of the key pathways that control responses to stresses including oxidative stress (Oeckinghaus & Ghosh, 2009; Hayden & Ghosh, 2012; Sivandzade et al, 2019). In addition, several EMT transcription factors (EMT-TFs) are direct targets of NF-κB (Huber et al, 2004; Lamouille et al, 2014; Dongre & Weinberg, 2019). For (ii), in response to oxidative stress, SIRT6 interacts with NRF2, a key transcriptional regulator of antioxidant defense, and up-regulates antioxidant genes by acting as a coactivator of NRF2 in human mesenchymal stem cells (Pan et al, 2016). For (iii), under oxidative stress, SIRT6 is recruited to the sites of DNA double-strand breaks (DSBs) and stimulates DSB repair by mono-ADP ribosylation of poly(ADP-ribose) polymerase 1 (PARP1) (Mao et al, 2011). SIRT6 also regulates base excision repair in a PARP1-dependent manner (Xu et al, 2015). Interestingly, SIRT6 activity to promote DSB repair is correlated with the lifespan of rodent species (Tian et al, 2019).

Here, we characterized RPE changes caused by NaIO₃-induced oxidative stress in mice. To assess RPE damage, we developed a method to measure each area of the three regions with distinct RPE morphologies (periphery, transitional zone, and center) using RPE flat-mounts. RPE cells in the transitional zone exhibited EMT-like molecular characteristics with increased EMT markers. RPE cells in the center were more susceptible to stresses than those in the periphery. In response to NaIO₃, SIRT6 rapidly translocated from the nucleus to the cytoplasm and colocalized with G3BP1, a marker of stress granules (SGs), leading to nuclear SIRT6 depletion particularly in the center. To overcome this SIRT6 depletion, we generated a transgenic mouse line with inducible SIRT6 overexpression in the nucleus of RPE. SIRT6 overexpression protected the RPE from NaIO₃ and partially preserved the expression of catalase, an anti-oxidant enzyme. These results show the topological differences of mouse RPE and suggest SIRT6 as a potential target for protecting RPE against oxidative stress-related damage and diseases, possibly including AMD.

# Results

## RPE changes caused by NaIO₃-induced oxidative stress in mice are divided into three regions with distinct morphologies

NaIO₃ has been widely used as a model of in vivo oxidative stress that causes RPE death followed by retinal degeneration in various mammals for decades. However, most of these studies focused on the retina, but not RPE damage itself, and only recently, more attention has been paid to the effects of NaIO₃ on the RPE (Kim et al, 2008; Zhao et al, 2011; Yang et al, 2018, 2021; Tang et al, 2021; Zhang et al, 2021). Therefore, we wanted to fill this knowledge gap and began the study by analyzing RPE morphological changes induced by NaIO₃ in mice. Using RPE/choroid flat-mounts (called RPE flat-mounts) with immunofluorescence of zonula occludens-1 (ZO-1; also known as tight junction protein 1, TJP1), which is located at the cell border and therefore outlines the shape of RPE cells, we conducted preliminary studies by observing RPE morphology at different time points from day 0–3 mo after NaIO₃ injection. The RPE was fragile and easily broken when making RPE flat-mounts before day 3. The transitional zone was not present on days 1 and 2 but appeared by day 3. The RPE structure became more stable on days 5–7, when making RPE flat-mounts was easier. We followed RPE damage after day 7, and the proportion of the three regions did not significantly change, at least up to 3 mo. Therefore, the earliest time point to analyze RPE flat-mounts with ease and consistency was day 7, which we chose as the universal time point for analyzing the extent of RPE damage throughout this study. During these preliminary studies, we also observed that RPE damage by NaIO₃ was not evenly distributed as previously noted (Xia et al, 2011; Ma et al, 2020; Upadhyay et al, 2020; Wolk et al, 2020; Yang et al, 2021; Zhang et al, 2021). Although RPE in the periphery was well preserved, RPE in the center (posterior) around the optic nerve head was severely damaged, either degenerated or completely lost, compared with control mice with the vehicle (Fig 1a and b). In addition, we consistently observed elongated RPE cells between normal-shaped RPE in the periphery and damaged RPE in the center (Fig 1c and d). Thus, we confirmed that RPE changes caused by NaIO₃ could be divided into three categories (regions): normal cobblestone-like RPE (periphery), elongated and enlarged RPE (transitional zone), and severely damaged or lost RPE (center) (A, B, and C, respectively, in Fig 1e). In a separate study, we also noticed that a higher dose of NaIO₃ caused a larger area of severely damaged RPE, and a lower dose of NaIO₃ resulted in a larger area of normal-appearing RPE (Yang et al, 2021), suggesting that these three regions might represent the different degrees of RPE damage, that is, no or mild damage in the periphery, moderate damage in the transitional zone, and severe damage in the center. In this separate study, we found that although NaIO₃ at 10 mg/kg body weight (BW) did not produce RPE morphological damage, NaIO₃ at 20 mg/kg BW caused severe RPE damage in the nearly entire RPE in male C57BL/6J mice (Yang et al, 2021). Because NaIO₃ at 15 mg/kg BW was between these two doses and caused severe RPE damage in roughly a half of the RPE, we assumed that this dose was more sensitive to modulating conditions and therefore used it in most of the experiments in the present study.

Based on these initial observations, we hypothesized that if we could quantify the three regions, we would be able to assess the severity of RPE damage overall. Therefore, we developed a morphometric method to quantify these three regions (A, B, and C) on RPE flat-mounts with ZO-1 immunofluorescence. Once each mouse was marked by ear tags, mice were identified only by ear tag numbers without referring to experimental conditions, which could make the process blind. After RPE flat-mounts were stained by ZO-1

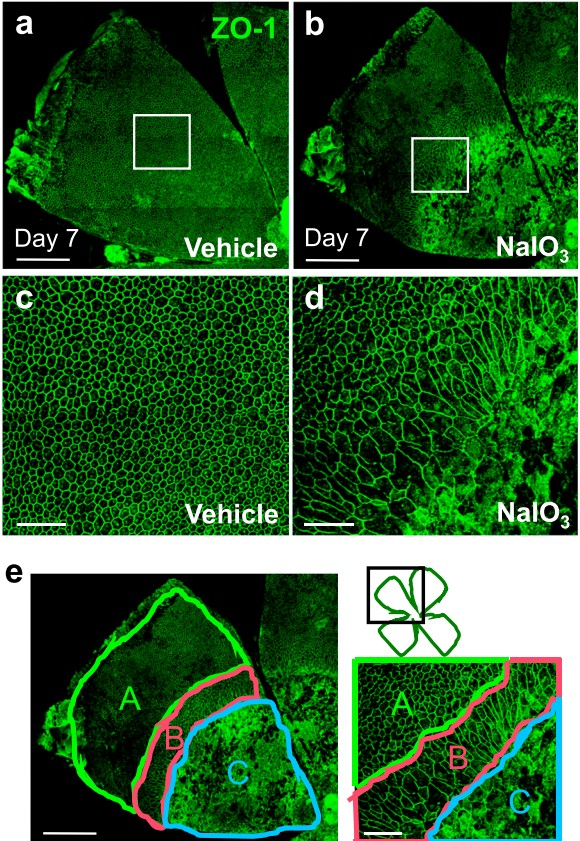

**Figure 1. NaIO₃-induced oxidative stress causes RPE damage with three distinct morphologies in mice.**
**(a, b, c, d, e)** RPE flat-mounts with immunofluorescence for zonula occludens-1 (ZO-1) (green). Mice were injected with PBS (vehicle) (a, c) or NaIO₃ (15 mg/kg body weight, BW) (b, d, e) on day 0 and analyzed on day 7. **(a, b)** One petal consisting of a quarter of the whole RPE flat-mounts; **(c, d)** higher magnification images of the white squared region in (a, b). RPE damage by NaIO₃ was prominent in the center around the optic nerve head, and elongated RPE cells were consistently observed between the periphery and the center. **(e)** RPE changes caused by NaIO₃ were divided into three categories (regions): normal cobblestone-like RPE (A, periphery), elongated and enlarged RPE (B, transitional zone), and damaged or lost RPE (C, center). A method to quantify these areas (A, B, C) was developed by measuring the number of pixels in each area using ImageJ and calculating the proportion (%) of each area to the entire RPE on flat-mounts. The detailed quantification methods and the criteria for separating these three regions are described in the Results section. Scale bars in the images: (a, b, e left) 500 *μm*; (c, d, e right) 100 *μm*.

immunofluorescence, an image of the entire flat-mounts was acquired by the tiling function of a confocal microscope. Then, the image was analyzed using the ImageJ software (US National Institutes of Health [NIH]) by two people, one who obtained the image and the other who was not involved in the process, and the two results were averaged. We used "aspect ratio (AR)" as the criteria for separating RPE between the periphery with normal-appearing cells and the transitional zone with elongated cells. AR is the ratio of the cell's major axis to the minor axis and reflects one of the cell's morphological characteristics (Kim et al, 2021; Ortolan et al, 2022). RPE cells with compact cobblestone appearance have AR closer to 1, and elongated cells have AR above 1, particularly higher than 1.5. Therefore, we arbitrarily separated the two regions using AR = 1.5 as

a cut-off. However, this separation was done by manual inspection, not by a computer-based format, and therefore it was approximate. The criteria for separating the central region with severely damaged or lost RPE were based on the observation that RPE was no longer recognizable as cells with the ZO-1-stained cell border. We included all RPE flat-mounts for analysis as far as we could recognize RPE cells with ZO-1 staining anywhere. After separating the three regions as described above, we measured each area by the number of pixels using ImageJ and calculated the proportion (%) of each region to the entire RPE. These initial findings prompted us to further characterize RPE changes and to find potential remedies for RPE damage.

### NaIO₃-induced oxidative stress leads to down-regulation of RPE markers and up-regulation of NF-κB targets in mouse RPE

We analyzed gene expression after NaIO₃ injection at mRNA and protein levels and the localization of selected proteins. We were interested in the transitional zone and therefore chose day 3 after NaIO₃ injection for analyzing protein localization because the transitional zone became clearly seen on day 3 onward. As RPE markers, we chose SOX9 and OTX2, two transcription factors important for RPE development and functions, because they are nuclear proteins and therefore it is easier to detect and count positive cells. We selected images on day 3 from the areas containing the junction of the three regions, periphery (labeled as Peri), transitional zone (Trans), and center (Cent), and these regions are demarcated by dotted lines (Fig 2a and b). Although both RPE markers were maintained in the periphery with normal-shaped RPE on day 3, they were no longer detectable by immunofluorescence in either the transitional zone or the center by day 3 (Fig 2a and b). We counted the number of SOX9- and OTX2-positive nuclei along with DAPI stained nuclei in the same area on day 3 and calculated the ratio (proportion, %) of positive nuclei for each factor to the DAPI stained nuclei from three independent experiments (Fig 2c). For the periphery, we used areas from the mid-far periphery in the original images for this quantification, not areas near the transitional zone in the images shown. The quantification results confirmed our observations described above. The mRNA levels of *Sox9* and *Otx2* analyzed by reverse transcription-quantitative PCR (RT–qPCR) also significantly decreased after NaIO₃ injection (Fig 2d). However, a more rapid and drastic reduction in the mRNA levels occurred with genes related to the visual cycle, one of the most RPE-specialized functions, including *Rpe65*, *Lrat*, *Rgr*, *Rdh5*, and *Rlbp1* (Fig 2d). Because NF-κB signaling is involved in the response to various stresses, we also tested the mRNA levels of NF-κB target genes, *Icam1*, *Fn1*, *Fth1*, *Irf1*, *Fas*, and *Ifnb1* (Fig 2e). Although these genes showed different patterns of changes, they were significantly up-regulated at either 24 h or day 7 or both after NaIO₃ injection at 15 mg/kg BW except *Ifnb1*. Protein levels were analyzed by Western blotting up to day 7 including day 3 in three separate gels: Gel 1 (Fig 2f, left panel) and Gels 2 and 3 (Fig S1a), and the quantification results of Western blot bands were consistent with those of immunofluorescence and RT–qPCR with some variabilities (Fig 2f, right 4 panels). These results show that NaIO₃-induced oxidative stress causes molecular changes of both RPE markers and NF-κB

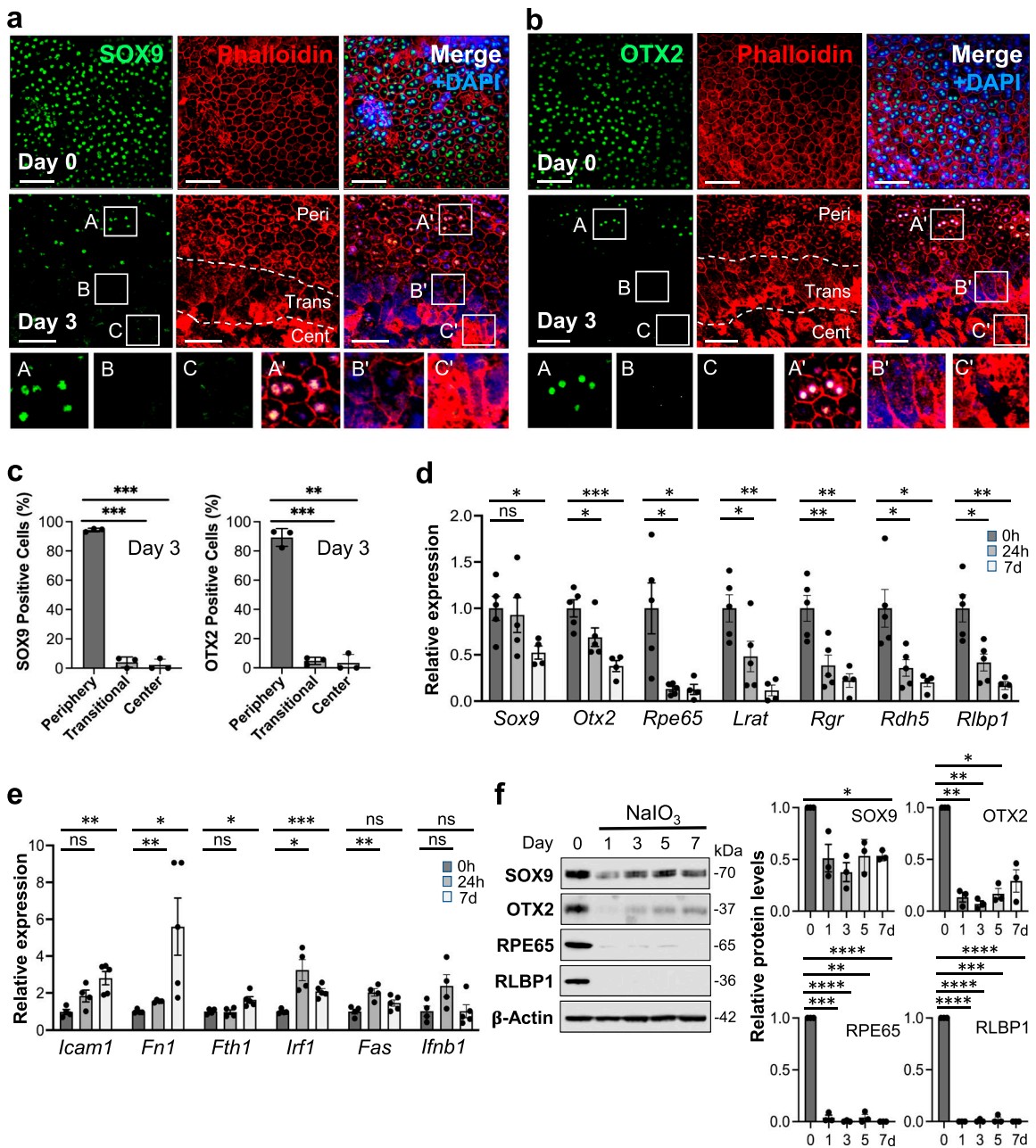

**Figure 2. NaIO₃-induced oxidative stress leads to down-regulation of RPE markers and up-regulation of NF-κB targets in mouse RPE.**
**(a, b)** Immunofluorescence of mouse RPE flat-mounts. Mice were injected with NaIO₃ (15 mg/kg BW) on day 0, and (a) SOX9 (green) and (b) OTX2 (green) were analyzed along with phalloidin (red) and nuclear stain DAPI (blue) on days 0 and 3. Representative images are shown. Images of day 3 are from the areas containing the junction of the three regions, periphery (labeled as Peri), transitional zone (Trans), and center (Cent), which are defined in Fig 1, and these regions are demarcated by dotted lines. We used aspect ratio AR ≥ 1.5 as the criteria for separating elongated cells in the transitional zone from normal cobblestone-like RPE in the periphery. Higher magnification images of the boxed areas (A, B, C and A′, B′, C′) are shown at the bottom. A′, B′, and C′ in the merged images correspond to A, B, and C in the SOX9 or OTX2 image, respectively. **(c)** The proportion of SOX9- and OTX2-positive cells in the three regions. The number of SOX9-positive nuclei and DAPI-stained nuclei was counted individually within the same area in each region on day 3, and the ratio of SOX9-positive nuclei to DAPI-stained nuclei was calculated and presented as SOX9-positive cells (%) (left panel). OTX2-positive cells (%) were calculated and presented in the same manner (right panel). Results are presented as mean ± SD (n = 3 mice for each region). Statistical significance was analyzed by one-way ANOVA. **(d)** The mRNA expression of RPE markers. Mice were injected with NaIO₃ (15 mg/kg BW) on day 0 (0 h), total RNA from mouse RPE without the choroid was prepared at 0 h, 24 h, and 7 d (day 7), and mRNA expression was analyzed by RT–qPCR. The mRNA levels were calculated using the $2^{-\Delta\Delta Ct}$ method with a geometric mean of three reference genes, *Gapdh*, *Hprt1*, and *Actb1*, for normalization. Relative expression is presented as the ratio to the mRNA level at 0 h. Results are presented as mean ± SEM with all individual data points (n = 4–5 mice for each time point). Statistical significance was analyzed by *t* test. **(e)** The mRNA expression of NF-κB target genes. The same samples used in (d) were analyzed by RT–qPCR, and results were calculated and presented in the same manner as described in (d). **(f)** The protein expression of RPE markers. Mice were injected with NaIO₃ (15 mg/kg BW) on day 0 and RPE protein lysates were prepared on days 0, 1, 3, 5, and 7. The protein levels were analyzed by Western blotting with antibodies against RPE markers indicated and β-actin for loading control, and the signal intensity of each band was quantified using ImageJ. The signal intensity of each protein was normalized by that of β-actin (protein/β-actin), and relative protein levels

targets before morphological changes become easily detectable after day 3.

### RPE changes caused by NaIO₃ show EMT-like characteristics in the transitional zone

As described above, elongated and enlarged RPE cells in the transitional zone were consistently observed by day 3 after NaIO₃ injection, and therefore, we chose day 3 for analyzing RPE morphological changes with immunofluorescence of EMT-related proteins (Fig 1b and d). Because RPE cells have been known to undergo EMT in some conditions, we hypothesized that these elongated RPE cells represent partial EMT caused by oxidative stress. SNAI1 (Snail), one of the key EMT-TFs, was barely detectable at the basal level (day 0) but became prominently expressed in the transitional zone by day 3 after NaIO₃ injection (15 mg/kg BW) (Fig 3a). Importantly, SNAI1 was also detected in normal-appearing RPE in the periphery on day 3 (Fig 3a). ZEB1, another EMT-TF, also increased by day 3 after NaIO₃ injection in the periphery with normal-shaped RPE cells, but ZEB1 was intensely stained, particularly in elongated cells in the transitional zone (Fig 3b). We quantified the signal intensity of SNAI1 and ZEB1 in the DAPI-stained nuclei in the three regions using ImageJ (20 nuclei for each region) from the day 3 images (Fig 3c). The quantification results confirm that both SNAI1 and ZEB1 were expressed at higher levels in the transitional zone than in the periphery or the center.

Next, we analyzed the mRNA levels of several EMT markers using RT–qPCR. *Snai1* was significantly up-regulated by 24 h after NaIO₃ injection (15 mg/kg BW), with its elevated level remaining on day 7 (Fig 3d). We could not detect a significant up-regulation of *Zeb1* as seen for ZEB1 in the transitional zone (Fig 3b), which is likely because of the limited area of ZEB1 up-regulation and different time points analyzed. In contrast, the mRNA level of *Vim* (vimentin), an intermediate filament expressed in mesenchymal cells, increased through day 7, but we did not see the increase of mRNA expression of *Acta2* (alpha-smooth muscle actin, α-SMA), an actin filament involved in cell motility and a marker of myofibroblasts (Fig 3d). In EMT, epithelial markers decrease in parallel with increased mesenchymal markers; therefore, we analyzed the mRNA levels of cadherins, *Cdh1* (E-cadherin), *Cdh2* (N-cadherin), and *Cdh3* (P-cadherin), and *Tjp1* (ZO-1). Although none of these genes showed significant changes in the mRNA levels, there was a trend of *Cdh3* mRNA gradually decreasing with time (Fig 3e). We recently found that P-cadherin is the highly dominant cadherin in the RPE in vivo, which is different from most of other epithelial cells where E-cadherin is the major subtype (Yang et al, 2018). Therefore, we further analyzed cadherin mRNA levels by quantifying the absolute amount of cDNAs generated from mRNAs in the same samples analyzed by RT–qPCR. Based on the standard curves, we calculated the amount of cDNA produced from 200 ng of total RNA as 52.8, 61.3, and 272 zmole at 0 h, 57.3, 57.1, and 228 zmol at 24 h, and 42.9, 45.6, and 134

zmole on day 7 for *Cdh1*, *Cdh2*, and *Cdh3*, respectively (Fig 3f). As control for the quantity and quality of RNA, we confirmed that these RPE RNA samples had the equivalent levels of *Gapdh* expression. Assuming that the efficiency of RT and qPCR was the same for all samples, *Cdh3* was significantly down-regulated by day 7, with *Cdh1* and *Cdh2* expressions remaining at similarly low levels.

We also analyzed protein levels of EMT markers by Western blotting up to day 7 including day 3 in three separate gels: Gel 1 (Fig 3g, top panel) and Gels 2 and 3 (Fig S1b), and the signal intensity of Western blot bands was quantified by ImageJ using β-actin as loading control. The results showed no significant differences in protein levels of SNAI1, VIM, and β-catenin in the whole RPE lysates during days 0–7, whereas α-SMA modestly increased on day 7 (Fig 3g, bottom 4 panels). It was conflicting that vimentin increased at the mRNA level on day 7, but its protein level was unchanged. We speculated that analyses of whole RPE lysates might not be the best approach because of regional differences of RPE damage and gene expression. Therefore, we performed immunofluorescence for VIM with RPE flat-mounts on day 7 after NaIO₃ injection to detect localized differences of VIM expression. Indeed, VIM was strongly stained in elongated cells in the transitional zone and in the center (Fig 3h). The VIM signal intensity was quantified inside the defined areas (36, 35, and 15 squares for the periphery, transitional zone, and center, respectively) using ImageJ (Fig S2). VIM protein levels were significantly higher in the transitional zone compared with the periphery where VIM was barely detectable (Fig 3i). These results indicate that oxidative stress induced by NaIO₃ elicits an EMT-like response in the transitional zone, and that some EMT molecular changes occur even in the periphery with normal-shaped RPE cells.

### NaIO₃ causes retinal photoreceptor cell death that is moderate in the retina facing the RPE transitional zone and extensive in the center

To examine how the RPE changes correlated with photoreceptor cell death, we used TUNEL staining of retinal sections, together with immunofluorescence of SOX9 as a marker for normal RPE and ZEB1 as a marker for the transitional zone. In our preliminary experiments, TUNEL signals in the outer nuclear layer (ONL) consisting of photoreceptor nuclei peaked on day 3 after NaIO₃ injection compared with those on days 1, 2, and 4, and therefore, we chose day 3 for our analyses. Comparing retinal sections with DAPI nuclear stain between days 0 and 3 after NaIO₃ injection, the outer nuclear layer was visibly thinner on day 3 (Fig 4a), particularly in the center where TUNEL staining was intense (Fig 4a and b). Concomitantly, SOX9 that was detected in the RPE on day 0 was no longer detectable on day 3 (Fig 4b). However, SOX9 in the inner nuclear layer (INL), where Muller glia expresses SOX9, was strongly and widely detected on day 3, suggesting that Muller glia is activated by oxidative stress and resultant retinal degeneration. In contrast, the retina was preserved at the far periphery (Fig 4a and c). We observed a clear border

were calculated as the ratio of protein/β-actin on different days to that on day 0. Samples from three independent experiments were analyzed by three separate gels, Gel 1 (left panel) and Gels 2 and 3 (Fig S1a). Results are presented as mean ± SEM with individual data points from the three gels (n = 3 mice for each time point) (right 4 panels). Statistical significance was analyzed by one-way ANOVA, and significance was marked in the graphs. Statistical significance throughout the figure: ns (not significant), *P < 0.05, **P < 0.01, ***P < 0.001, and ****P < 0.0001. Scale bars in the images: (a, b) 50 μm.

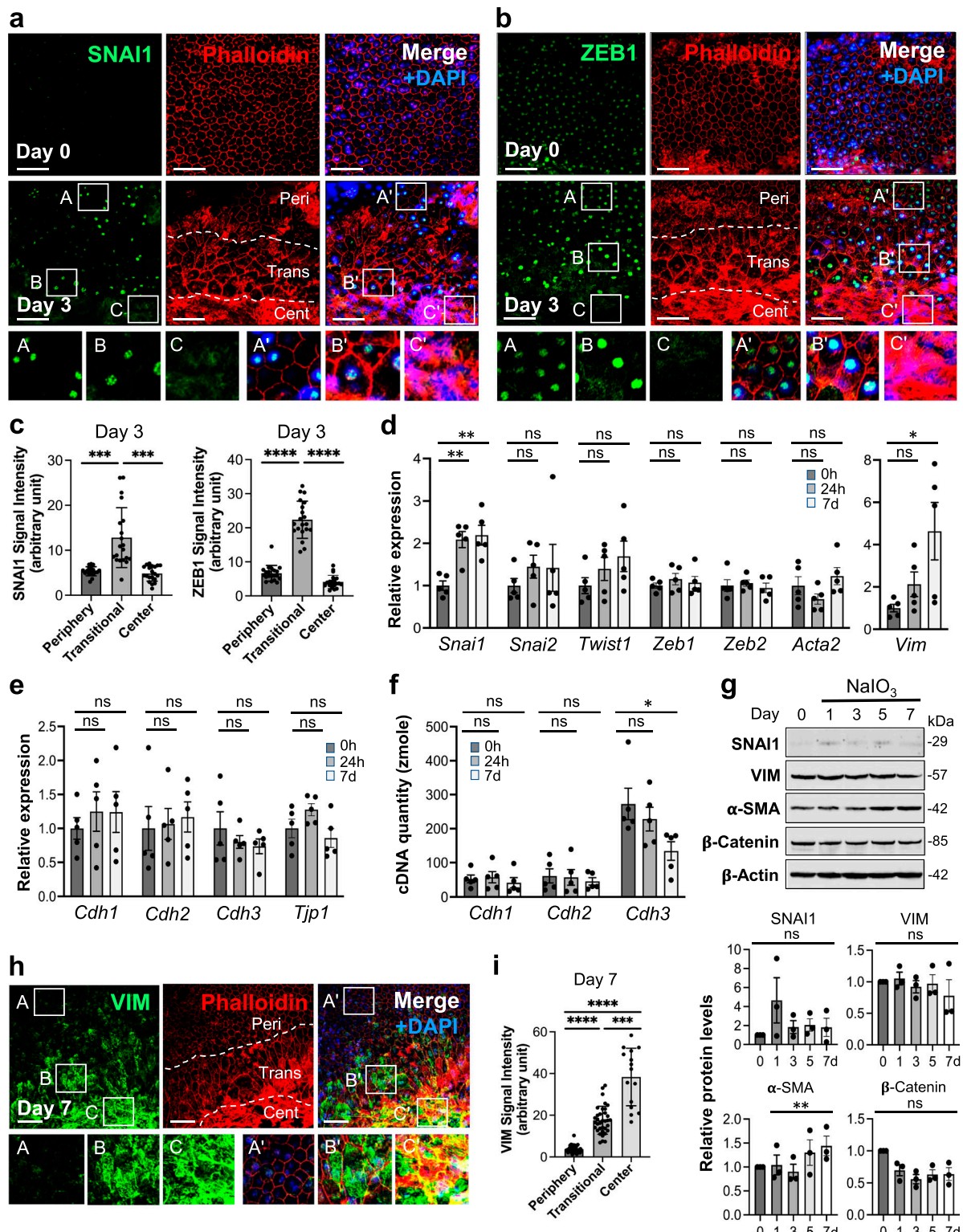

**Figure 3.  RPE damage caused by NaIO₃ shows EMT-like characteristics.**
**(a, b)** Immunofluorescence of mouse RPE flat-mounts. Mice were injected with NaIO₃ (15 mg/kg BW) on day 0, and (a) SNAI1 (green) and (b) ZEB1 (green) were analyzed along with phalloidin (red) and nuclear stain DAPI (blue) on days 0 and 3. Representative images are shown. Images on day 3 are from the areas including the junction of the three regions, periphery (labeled as Peri), transitional zone (Trans), and center (Cent), which are defined in Fig 1, and these regions are demarcated by dotted lines. Aspect ratio AR ≥ 1.5 was used as the criteria for separating elongated cells in the transitional zone from normal cobblestone-like RPE in the periphery. Higher magnification images of the boxed areas (A, B, C and A′, B′, C′) are shown at the bottom. A′, B′, and C′ in the merged images correspond to A, B, and C in the SNAI1 or ZEB1 image, respectively. **(c)** Quantification of signal intensity of SNAI1 and ZEB1 immunofluorescence in the three regions. The signal intensity of SNAI1 in the DAPI-stained

between TUNEL negative and positive areas in the retina, which coincided with a border between SOX9-positive and -negative RPE layers (Fig 4c, *arrowhead*). Using ZEB1 to mark the transitional zone, we found that this border area corresponded to the transitional zone with up-regulated ZEB1 (Fig 4d, *arrowhead*), where TUNEL was positive but appeared weaker than in the center. To confirm our impression from the images that TUNEL staining was more sporadic in the transitional zone, we quantified the signal intensity of TUNEL in the squared areas in the enlarged images using ImageJ (Fig 4e and f). The squared areas are the same in size and partially overlap the next squares at both sides and their horizontal positions are indicated by numbered lines at the top, 1–3 in the center and 4–12 in the periphery. Although the squares were slid from left to right at the same level, the numbered lines are placed at different vertical positions because of the limited space. The images of the center and periphery are from the same set of experiments. The total TUNEL signals were lower in the transitional zone (locations 6–8) in both experiments (Fig 4e and f). These results indicate that the extent of photoreceptor death correlates with that of RPE damage by $NaIO_3$ in that photoreceptor death is moderate in the retina facing the transitional zone and extensive in the center, and that TUNEL staining of the retina can demarcate the three regions with distinct RPE damage. These results seem to support the hypothesis that photoreceptor death is mostly secondary to RPE death or dysfunction in $NaIO_3$-injected animals.

### Mouse RPE in the center is more susceptible to oxidative stress than RPE in the periphery

The results obtained thus far indicated that the response of RPE cells in the center to oxidative stress was different from that in the periphery. Therefore, we further investigated this difference. Based on the reports that SIRT6 functions against oxidative stress as a coactivator of NRF2, a key regulator of antioxidant genes (Pan et al, 2016), and that SIRT6 also regulates the assembly of stress granules (SGs) in the cytoplasm in response to various stresses

(Jedrusik-Bode et al, 2013; Simeoni et al, 2013), we hypothesized that SIRT6 might behave differently in the peripheral and central RPE. In our analyses below, we defined "periphery" and "center" as follows. The entire RPE flat-mount was divided into three concentric zones from the periphery to the optic nerve head (center), with one-third of the length of a radial line for each zone (periphery, middle, and center). First, we analyzed SIRT6 distribution in mouse RPE 1 h after $NaIO_3$ injection at different doses (Fig 5a). Immunofluorescence of RPE flat-mounts showed that SIRT6 was dominantly localized in the nucleus in both the periphery and center with $NaIO_3$ at 0 and 5 mg/ kg BW. With $NaIO_3$ at 10 mg/kg BW, although SIRT6 was still dominantly localized in the nucleus in the periphery, its nuclear localization decreased in the center. Nuclear SIRT6 in the periphery began to decrease with $NaIO_3$ at 15 mg/kg BW, and it was lower but still easily detectable even at 20 mg/kg BW. In contrast, SIRT6 in the nucleus further decreased in the center and became barely detectable with $NaIO_3$ at 15 and 20 mg/kg BW (Fig 5a). To avoid saturation of fluorescence signals, we obtained these images using the same setting with low exposure, which is detailed in the figure legends. We quantified the intensity of SIRT6 signals in the DAPI-stained nuclei (20 nuclei for each condition) using ImageJ (Fig 5a, bottom panels). There was no statistically significant difference in SIRT6 signal intensity between $NaIO_3$ at 15 and 20 mg/kg BW in either the periphery or the center. We speculate that $NaIO_3$ at 15 mg/kg BW already exerts a near maximum effect leading to the loss of nuclear SIRT6. These results show that nuclear SIRT6 is lost with lower doses of $NaIO_3$ in central RPE compared with peripheral RPE.

Because the results described above were obtained in mice, we suspected two possible mechanisms for this topological difference in the RPE vulnerability to $NaIO_3$: (1) a gradient of $NaIO_3$ concentration created by the blood flow from center to periphery and (2) differences of intrinsic properties of central and peripheral RPE. To test the second scenario, we developed the ex vivo mouse RPE system by incubating the RPE/choroid/sclera eyecups in $CO_2$-independent medium at 37°C (Fig 5b and c). After 1 h incubation, nuclear SIRT6 in the center of ex vivo RPE significantly decreased with a concomitant increase of cytoplasmic SIRT6 compared with

nuclei was quantified using ImageJ (n = 20 nuclei for each region) from the day 3 image in (a) and presented as mean ± SD (left panel). The signal intensity of ZEB1 from the day 3 image in (b) was quantified and presented in the same manner (right panel). Statistical significance was analyzed by one-way ANOVA. **(d)** The mRNA expression of EMT markers. Mice were injected with $NaIO_3$ (15 mg/kg BW) on day 0 (0 h), total RNA from mouse RPE devoid of the choroid was extracted at 0 h, 24 h, and 7 d (day 7), and mRNA expression was analyzed by RT–qPCR. The mRNA levels were calculated using the $2^{-\Delta\Delta Ct}$ method with a geometric mean of three reference genes, *Gapdh*, *Hprt1*, and *Actb1*, for normalization. Relative expression is presented as the ratio to the mRNA level at 0 h. Results are presented as mean ± SEM with all individual data points (n = 5 mice for each time point). Statistical significance was analyzed by *t* test. **(e)** The mRNA expression of epithelial markers. The same samples used in (d) were analyzed by RT–qPCR, and results were calculated and presented in the same manner as described in (d). **(f)** Absolute quantification of cDNA to assess the mRNA quantity of *Cdh1*, *Cdh2*, and *Cdh3*. The same RNA samples used in (d) were analyzed by RT–qPCR, along with gel-purified PCR products to create a standard curve for each gene ranging from 1 attomole (amole) to 0.1 zeptomole (zmole). Based on Ct values of the standard curve, the cDNA quantity was calculated for 200 ng total RNA used for the initial cDNA synthesis. Results are presented as mean ± SEM with all individual data points (n = 5 mice for each time point). Statistical significance was analyzed by *t* test. **(g)** The protein expression of EMT-related factors. Mice were injected with $NaIO_3$ (15 mg/kg BW) on day 0 and RPE protein lysates were prepared on days 0, 1, 3, 5, and 7. The protein levels were analyzed by Western blotting with antibodies against EMT-related proteins indicated and control *β*-actin, and the signal intensity of each band was quantified using ImageJ. The signal intensity of each protein was normalized by that of *β*-actin (protein/*β*-actin), and relative protein levels were calculated as the ratio of protein/ *β*-actin on different days to that on day 0. Samples from three independent experiments were analyzed by three separate gels: Gel 1 (top panel) and Gels 2 and 3 (Fig S1b). Results are presented as mean ± SEM with individual data points from the three gels (n = 3 mice for each time point) (bottom 4 panels). Statistical significance was analyzed by one-way ANOVA, and no significance (ns) is marked only in the limited cases in the graphs. **(h)** VIM protein expression in mouse RPE. Mice were injected with $NaIO_3$ (15 mg/kg BW) on day 0, and the expression of VIM (green) was analyzed by immunofluorescence of mouse RPE flat-mounts along with phalloidin (red) and DAPI (blue) on day 7. Representative images are from the area including the junction of the three regions and presented in the same manner as in (a, b), with higher magnification images of the boxed areas shown at the bottom. **(i)** Quantification of signal intensity of VIM immunofluorescence in the three regions. The VIM signal intensity inside the defined areas in the images shown in (h) was quantified using ImageJ. These areas are defined by squares and shown in Fig S2. Results are presented as mean ± SD (n = 36, 35, and 15 squares for the periphery, transitional zone, and center, respectively). Statistical significance was analyzed by one-way ANOVA. Statistical significance throughout the figure: ns (not significant), *$P < 0.05$, **$P < 0.01$, ***$P < 0.001$, and ****$P < 0.0001$. Scale bars in the images: (a, b, h) 50 *μm*.

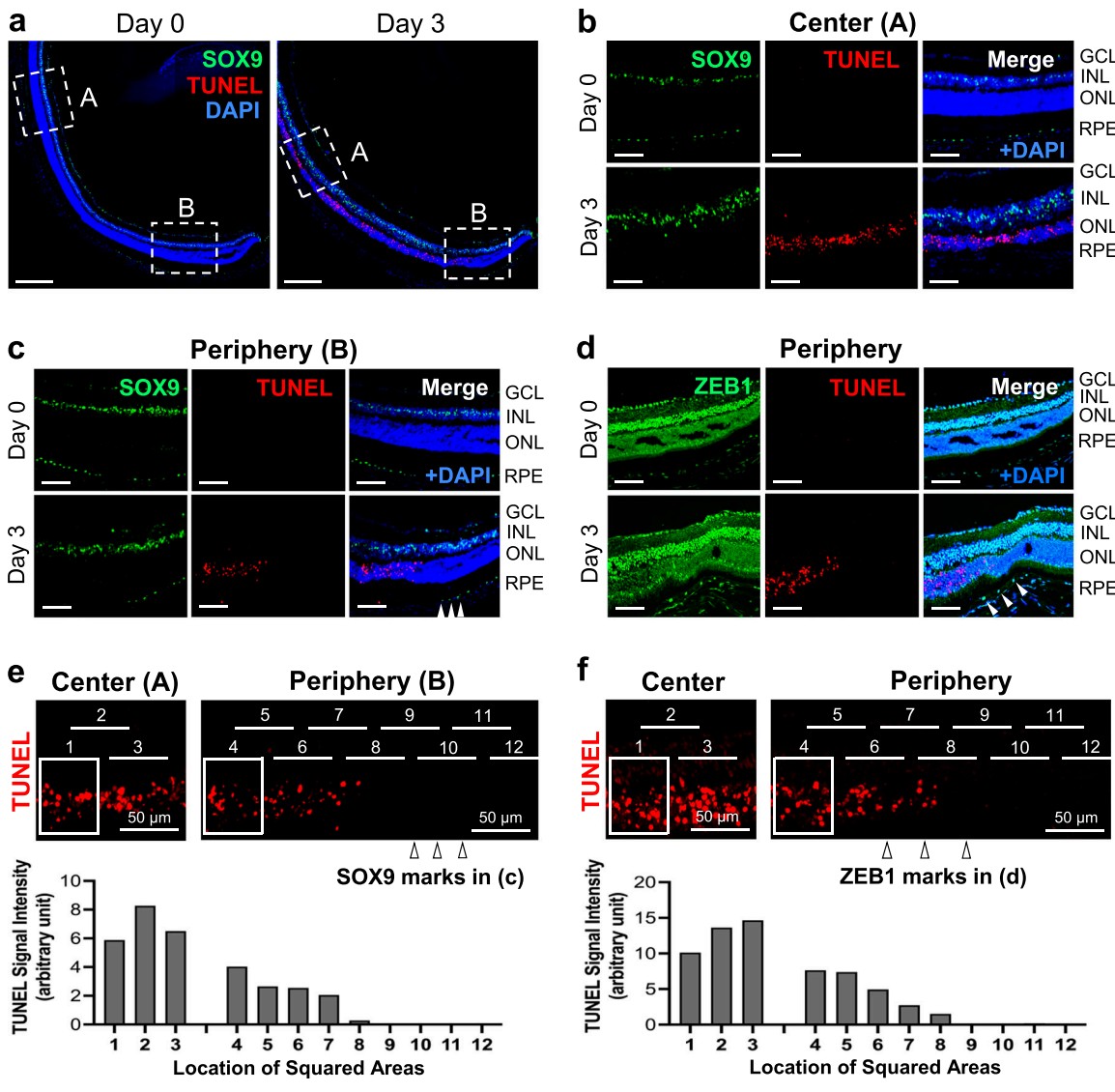

**Figure 4. NaIO$_3$ causes retinal photoreceptor cell death that is moderate in the retina facing the RPE transitional zone and extensive in the center.**
**(a, b, c, d)** Immunofluorescence of mouse retinal sections. Mice were injected with NaIO$_3$ (15 mg/kg BW) on day 0, and (a, b, c) SOX9 (green) and (d) ZEB1 (green) were analyzed together with TUNEL stain (red) and nuclear stain DAPI (blue) on days 0 and 3. Representative images are shown. **(a)** The images correspond to a half of the retina from the center (square A) to the periphery (square B). **(b)** Higher magnification images of square A in (a). RPE marker SOX9 was no longer detectable, and ONL was clearly thinner with intense TUNEL staining on day 3. **(c)** Higher magnification images of square B in (a). There was a clear border between TUNEL-negative and -positive ONL, which matched the border between SOX9-positive (*arrowhead*) and -negative RPE, and ONL was preserved at the far periphery on day 3. **(d)** Higher magnification images of the periphery from different retinal sections. As seen in (c), a clear border was present between TUNEL-negative and -positive ONL, which matched the transitional zone marked by ZEB1 up-regulation (*arrowhead*) on day 3. GCL, ganglion cell layer; INL, inner nuclear layer; ONL, outer nuclear layer. **(e, f)** Quantification of the signal intensity of TUNEL staining. The signal intensity in the squared areas in the enlarged images of TUNEL staining shown in (b, c, d) was quantified using ImageJ. These squared areas are same in size and partially overlap the next squares at both sides, and the horizontal position of these squares is indicated by numbered lines at the top (1–3 in the center and 4–12 in the periphery). Although these squares are located at the same level vertically, the numbered lines are placed at different vertical positions because of the limited space for marking. The images of the center and periphery are from the same set of experiments. Scale bars in the images: (a) 200 $\mu$m; (b, c, d, e, f) 50 $\mu$m.

fresh RPE (Fig 5b). To confirm this observation, we quantified the signal intensity of SIRT6 in both DAPI stained nuclei and cytoplasmic areas of the same size as nuclei (20 nuclei and 20 cytoplasmic areas for each condition) using ImageJ. Relative SIRT6 signals were calculated as the ratio of nuclear SIRT6 signals to the average of 20 cytoplasmic signals in each image (Fig 5b, bottom panel). The results show that the nucleus/cytoplasm ratio of SIRT6

signals was significantly lower in the center of ex vivo RPE. These ex vivo RPE cells were not exposed to oxidative stress reagents; however, peeling off the retina likely caused a physical stress to RPE cells whose apical microvilli tightly interdigitate with the outer segments of retinal photoreceptors. Such a physical stress would be one of the pathological consequences of retinal detachment. Using this ex vivo RPE system, we tested the response to oxidative

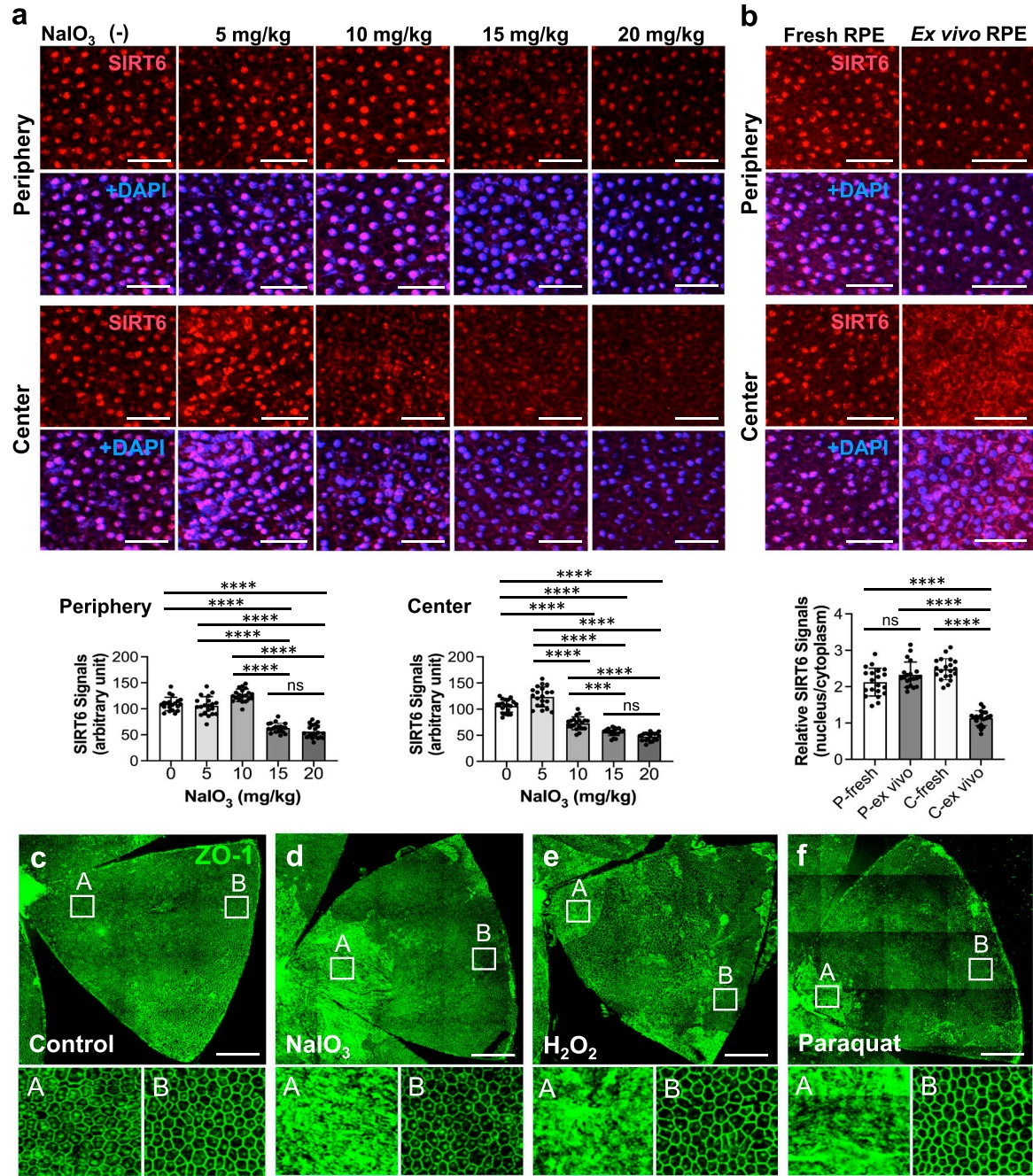

**Figure 5. Mouse RPE cells in the center are more susceptible to oxidative stress than those in the periphery.**

**(a)** Immunofluorescence of mouse RPE flat-mounts. Mice were injected with different doses of NaIO₃ (0, 5, 10, 15, and 20 mg/kg BW), and the distribution of SIRT6 (red) was analyzed 1 h later with nuclear stain DAPI (blue). All images were taken using the same low exposure setting (for SIRT6 [Alexa 546]: laser transmission, 80%; gain, 800/ 1,250; digital offset, −0.03). Representative images are shown. The signal intensity of SIRT6 in the DAPI-stained nuclei in the images was quantified using ImageJ (n = 20 nuclei for each NaIO₃ dose) and presented as mean ± SD (bottom panels). **(b)** Immunofluorescence of flat-mounts of ex vivo mouse RPE. After peeling off the retina, the RPE/choroid/sclera eyecup was incubated in CO₂-independent medium at 37°C as ex vivo mouse RPE. After 1 h incubation, SIRT6 (red) was stained along with DAPI (blue). Representative images are shown. The signal intensity of SIRT6 was quantified using ImageJ both in the DAPI-stained nuclei and in the cytoplasmic areas of the same size as the nuclei (n = 20 nuclei and 20 cytoplasmic areas for each condition). Relative SIRT6 signals were calculated as the ratio of nuclear SIRT6 signals to the average of 20 cytoplasmic signals and presented as mean ± SD (bottom panel). Analyzed were the periphery of fresh RPE (labeled as P-fresh) and ex vivo RPE (P-ex vivo) and the center of fresh RPE (C-fresh) and ex vivo RPE (C-ex vivo). **(c, d, e, f)** Immunofluorescence of flat-mounts of ex vivo mouse RPE. **(d, e, f)** The RPE/choroid/sclera eyecup was incubated with a low dose of three reagents inducing oxidative stress, (d) NaIO₃ (7,500 μg/ml, 1 h), (e) H₂O₂ (10 mM, 3 h), and (f) paraquat (2 mM, 3 h), followed by immunofluorescence for ZO-1 (green). Representative images are shown. Higher magnification images of the boxed areas, A (center) and B (periphery), are shown at the bottom. Statistical significance throughout the figure: ns (not significant), ***P < 0.001, and ****P < 0.0001 (one-way ANOVA). Not all ns are marked. Scale bars in the images: (a, b) 50 μm; (c, d, e, f) 500 μm.

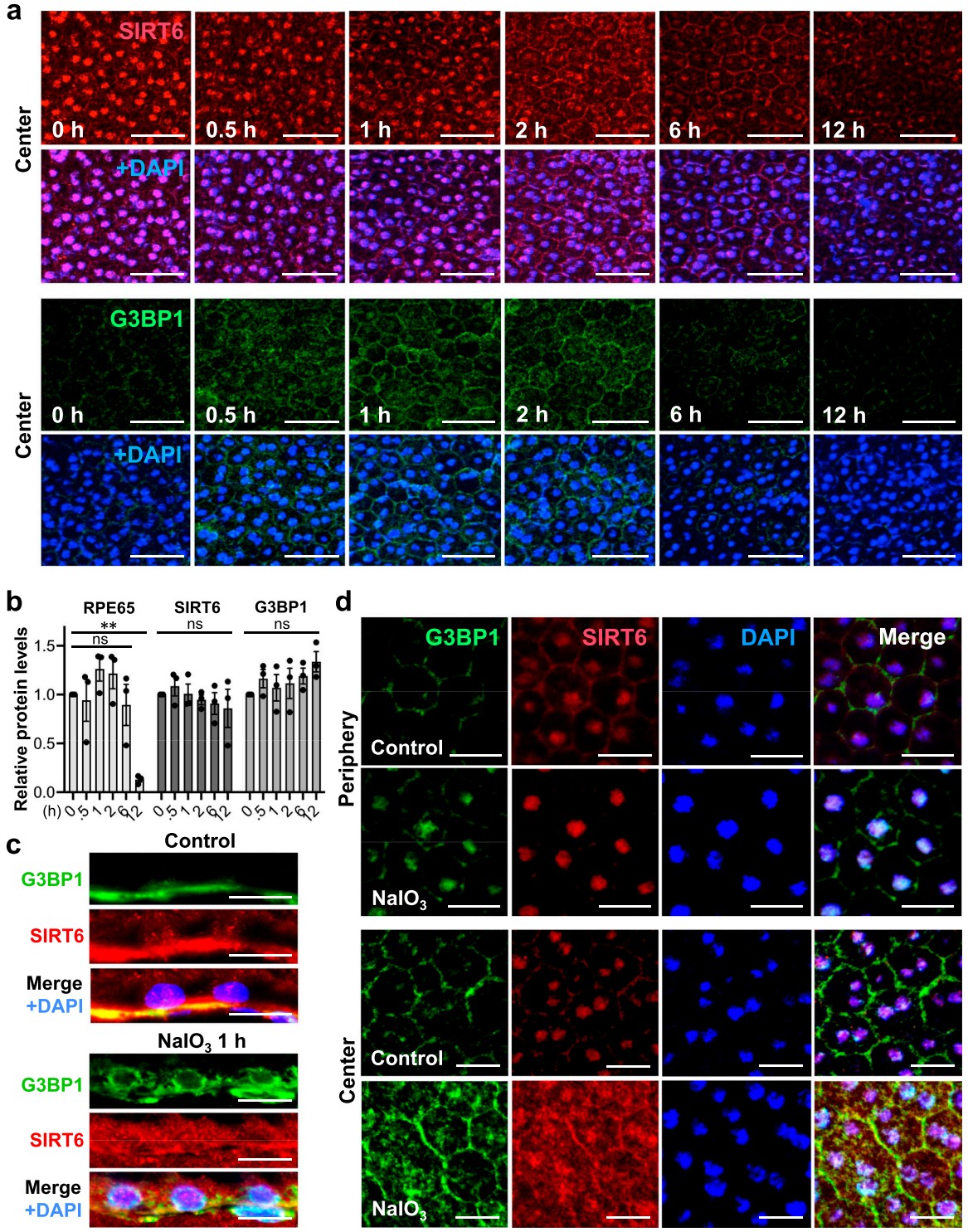

**Figure 6. In response to oxidative stress, SIRT6 rapidly translocates to the cytoplasm and colocalizes with G3BP1.**
**(a)** Immunofluorescence of mouse RPE flat-mounts. Mice were injected with $NaIO_3$ (20 mg/kg BW), and SIRT6 (red) and G3BP1 (green) were stained at different time points (0, 0.5, 1, 2, 6, and 12 h) along with DAPI (blue). All images were taken using the same low exposure setting (for SIRT6 [Alexa 546]: laser transmission, 80%; gain, 800/1,250; digital offset, −0.03). Representative images are shown. The signal intensity of SIRT6 in the nucleus was quantified and included in Fig S3. **(b)** The protein levels of SIRT6 and G3BP1. Mice were injected with $NaIO_3$ (20 mg/kg BW), and RPE protein lysates were prepared at the same time points as in (a). The protein levels were analyzed by Western blotting with antibodies against RPE65, SIRT6, G3BP1, and control $β$-actin, and the signal intensity of each band was quantified using ImageJ. The signal intensity of each protein was normalized by that of $β$-actin (protein/$β$-actin), and relative protein levels were calculated as the ratio of protein/$β$-actin at different time points to that at 0 h. Samples from three independent experiments were analyzed with three separate gels: Gels 1, 2, and 3 (Fig S1c). Results are presented as mean ± SEM with individual data points from the three gels (n = 3 mice for each time point). Statistical significance is shown by ns (not significant) and **$P < 0.01$ (one-way ANOVA),

stress induced by three reagents, NaIO$_3$ (7,500 $\mu$g/ml for 1 h), H$_2$O$_2$ (10 mM for 3 h), and paraquat (2 mM for 3 h) (Fig 5c–f). RPE damage caused by all three reagents at low doses that produce a localized damage occurred in the center, with peripheral RPE being preserved at least morphologically (Fig 5c–f, bottom panels). These results support the existence of intrinsic differences between central and peripheral RPE cells. However, the vascular mechanism creating different local concentrations of NaIO$_3$ is not mutually exclusive with RPE cell intrinsic differences, and we speculate that both mechanisms are at work.

### In response to oxidative stress, SIRT6 rapidly translocates to the cytoplasm and colocalizes with G3BP1

After observing the quick translocation of SIRT6 to the cytoplasm in response to oxidative stress in mouse RPE, we wanted to know whether SIRT6 was associated with the formation of stress granules (SGs) in this context. First, we analyzed SG formation at different time points (0, 0.5, 1, 2, 6, and 12 h) after NaIO$_3$ injection (20 mg/kg BW) using RPE flat-mounts with immunofluorescence for SIRT6 and G3BP1 (Fig 6a). G3BP1 is a key component of SGs and commonly used as a marker of SGs (Tourriere et al, 2003; Guillen-Boixet et al, 2020; Yang et al, 2020). We used NaIO$_3$ at 20 mg/kg BW in these experiments to give a stronger oxidative stress rather than 15 mg/kg BW to make certain that SIRT6 translocation could occur clearly. By 30 min after NaIO$_3$ injection, G3BP1 aggregates appeared in parallel with the decrease of nuclear SIRT6. Although G3BP1 staining returned to the baseline by 6 h, nuclear SIRT6 gradually decreased with time and became barely detectable without returning to the basal state in the time frame analyzed. The signal intensity of SIRT6 was quantified in the DAPI-stained nuclei (20 nuclei for each time point) by ImageJ, confirming the gradual decrease of nuclear SIRT6 (Fig S3). To rule out the possibility that SIRT6 reduction in the nucleus was because of protein degradation, we analyzed the protein levels in RPE lysates by Western blotting. Samples were analyzed with antibodies against RPE65, SIRT6, G3BP1, and control $\beta$-actin in three separate gels: Gels 1, 2, and 3 (Fig S1c), and the signal intensity of each band was quantified using ImageJ (Fig 6b). SIRT6 protein levels were unchanged during this time frame analyzed. In contrast, RPE65 decreased dramatically by 12 h, consistent with our earlier results that RPE65 was highly sensitive to oxidative stress and became nearly undetectable by 24 h after NaIO$_3$ injection (Fig 2f). Next, we tested whether SIRT6 and G3BP1 colocalized 1 h after NaIO$_3$ injection using immunofluorescence of retinal sections (Fig 6c) and RPE flat-mounts (Fig 6d). Based on the time-course analyses of SG formation described above, we chose the 1 h time point because it was within the period of active SG formation. At 1 h, SIRT6 was already massively present in the cytoplasm and colocalized with G3BP1 (yellow aggregates) in the center, but SIRT6 remained in the nucleus in the periphery (Fig 6d). These results show that SIRT6 rapidly translocates to the cytoplasm

and associates with G3BP1 in SG aggregates in response to oxidative stress in mouse RPE.

### SIRT6 is successfully induced and stays in the nucleus of RPE cells in transgenic mice

SIRT6 has multiple functions that are important for chromatin regulation, metabolism, DNA repair, antioxidant defense, antiaging, and longevity (Chang et al, 2020). Most of these functions are linked to SIRT6's roles in the nucleus. However, as we observed, SIRT6 quickly translocates to the cytoplasm of mouse RPE in response to oxidative stress. This phenomenon presents a challenge that SIRT6 is not available in the nucleus when it is needed. To overcome this problem and test the role of SIRT6 in the RPE, we generated transgenic mice with inducible SIRT6 overexpression in the nucleus of RPE cells (Fig 7a). For this transgenic construct, we used the human *BEST1* –585 to +38 bp (–585/+38) promoter (*BEST1* p) to direct expression in the RPE and designed an inducible SIRT6 expression by in-frame fusion of human SIRT6 coding sequence with the 4-hydroxytamoxifen-responsive mutant estrogen receptor ER$^{T2}$ (Fig 7a, top). In these mice, tamoxifen (Tmx) induces SIRT6 in the nucleus through the ligand-dependent dimerization and nuclear translocation of ER$^{T2}$ (Feil et al, 1997; Indra et al, 1999). To show the spatial characteristics of this human *BEST1* promoter in mouse RPE, an image of whole RPE flat-mount that was stained with X-gal for $\beta$-galactosidase (lacZ) from the *BEST1* –585/+38 promoter–*lacZ* transgenic mice (Esumi et al, 2004) is included (Fig 7a, bottom). This *BEST1* promoter is more active in the center than in the periphery where the staining is patchy. First, we wanted to confirm that SIRT6 was successfully induced in the nucleus of mouse RPE by Tmx using RPE flat-mounts with SIRT6 immunofluorescence (Fig 7b). SIRT6 transgenic mice (S4 line) were injected i.p. with Tmx (0.5 mg/mouse/day) or vehicle (10% ethanol in sunflower oil) for 3 days (days 1–3), and SIRT6 was analyzed on day 4 along with wild-type (Wt) mice. SIRT6 was induced with Tmx in the RPE nuclei more prominently in the center, where its leaky expression was also observed with vehicle in the transgenic mice compared with Wt mice. The distribution of induced SIRT6 was similar to the pattern of lacZ expression driven by the same promoter (Fig 7a and b). These results showed that SIRT6 was successfully induced with Tmx in our SIRT6 transgenic mice as designed. Next, we analyzed SIRT6 protein induction by Western blotting. SIRT6 transgenic mice (S4 line) were injected i.p. with Tmx or vehicle for 3 days, and RPE protein lysates were analyzed on day 4 in two separate gels, Gel 1 (Fig 7c, top panel) and Gel 2 (Fig S1d). Total samples from SIRT6 transgenic mice were three whole RPE lysates with vehicle (Oil, w), four whole RPE lysates with Tmx (Tmx, w), and five central RPE lysates with Tmx (Tmx, c), and the same whole RPE lysates from wild-type mice (Wt, w) were included in both gels for control. The signal intensity of each band was quantified using ImageJ and relative SIRT6 level in each sample was calculated as the ratio to the level in wild-type RPE (Wt, w)

without marking all ns in the graph. **(c)** Immunofluorescence of mouse retinal sections. Mice were injected with NaIO$_3$ (20 mg/kg BW), and SIRT6 (red) and G3BP1 (green) were stained along with DAPI (blue) at 1 h after NaIO$_3$ injection. Representative images are shown. **(d)** Immunofluorescence of mouse RPE flat-mounts. The same experimental conditions as described above in (c) were also used for RPE flat-mounts. Representative images are shown. Scale bars in the images: (a) 50 $\mu$m; (c) 10 $\mu$m; (d) 20 $\mu$m.

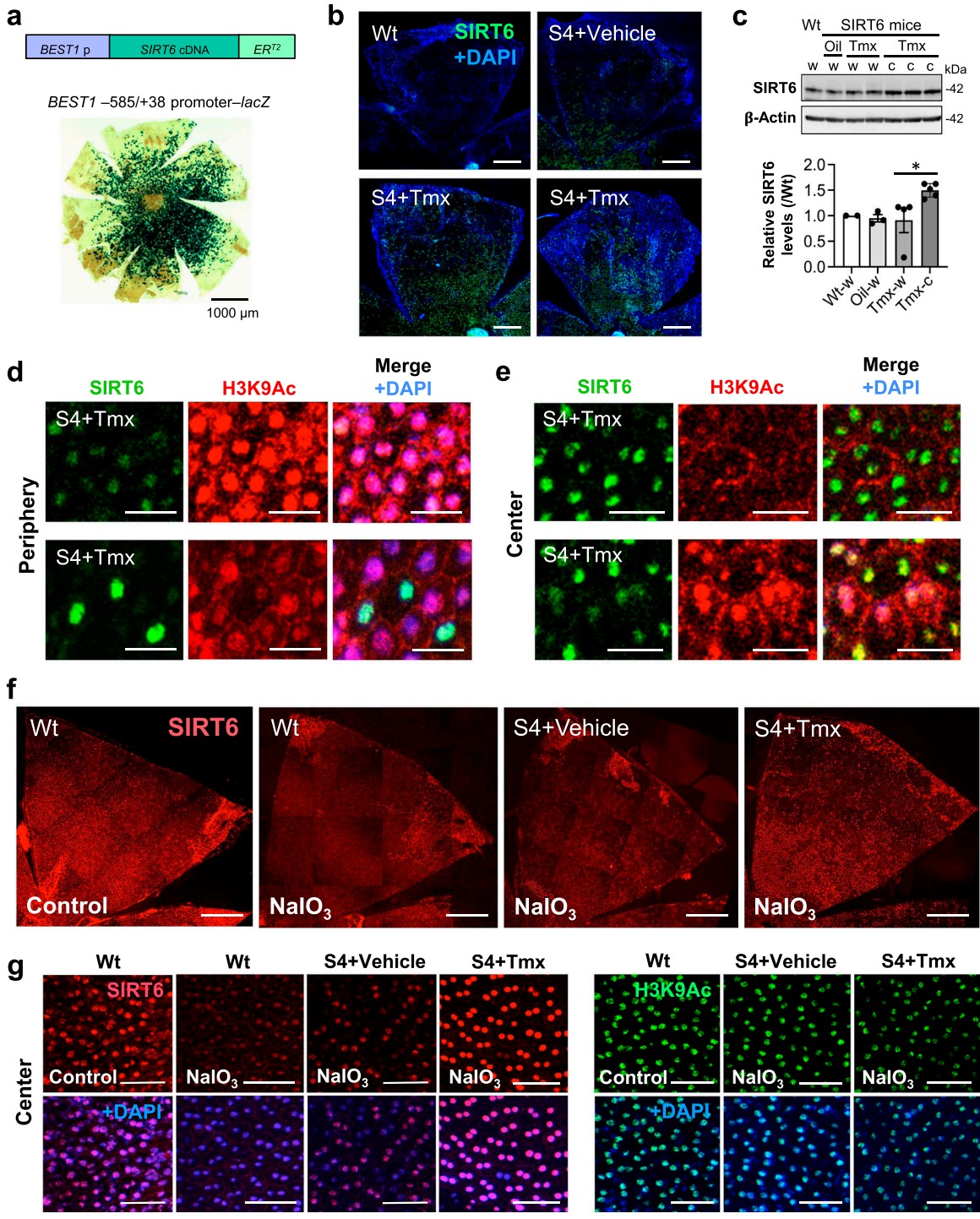

**Figure 7. SIRT6 is successfully induced and stays in the nucleus of RPE cells in transgenic mice.**

**(a)** Transgenic mouse construct for inducible SIRT6 overexpression in the RPE. The human *BEST1* −585 to +38 bp (−585/+38) promoter (*BEST1* p) was used to drive the expression in the RPE, and inducible SIRT6 expression was designed by in-frame fusion of human SIRT6 coding sequence with the 4-hydroxytamoxifen-responsive mutant estrogen receptor ER^T2, which is activated with tamoxifen (Tmx). A representative whole RPE flat-mount stained with X-gal for β-galactosidase (lacZ) from the *BEST1* −585/+38 promoter–*lacZ* transgenic mice shows the spatial characteristics of this promoter in mouse RPE (bottom). **(b)** Immunofluorescence of mouse RPE flat-mounts. SIRT6 transgenic mice (S4 line) were injected i.p. with Tmx (0.5 mg/mouse/day) or vehicle (10% ethanol in sunflower oil) for 3 days (days 1–3), and SIRT6 (green) was stained together with nuclear stain DAPI (blue) on day 4. All images were taken using the same microscopic settings. Representative images are shown for each condition along with Wt mice. **(c)** SIRT6 protein levels induced by Tmx. SIRT6 transgenic mice (S4 line) were injected i.p. with Tmx or vehicle for 3 days (days 1–3), and RPE protein lysates were analyzed on day 4 by Western blotting with antibodies against SIRT6 and control β-actin. Samples consisting of 3 whole RPE lysates from mice with vehicle (Oil, w), 4 whole RPE lysates from mice with Tmx (Tmx, w), and 5 central RPE lysates from mice with Tmx (Tmx, c) were analyzed in two separate gels, Gel 1 (top panel) and Gel 2 (Fig S1d), using the same wild-type whole RPE lysates (Wt, w) in both gels for control. The signal intensity of each band was quantified using ImageJ. SIRT6 signals were

(Fig 7c, bottom panel). Although SIRT6 induction with Tmx was not detectable in the whole RPE lysates, SIRT6 levels with Tmx were higher in the central RPE compared with the whole RPE, consistent with the feature of the *BEST1* promoter that is more active in the center (Esumi et al, 2004, 2007). Because our anti-SIRT6 antibody reacts with both mouse and human SIRT6, we speculate that mouse SIRT6 likely masked the induction of human SIRT6 to some extent. Next, we wanted to confirm that SIRT6 induction could reduce acetylation of H3K9 (H3K9Ac), a target of SIRT6's deacetylase function. SIRT6 transgenic mice (S4 line) were treated with Tmx or vehicle for 3 days, and SIRT6 (green) and H3K9Ac (red) were analyzed on day 4 using immunofluorescence of mouse RPE flat-mounts. Representative images from the periphery and center (Fig 7d and e, respectively, upper panels) and specific areas with sporadic SIRT6 induction in the periphery and poor SIRT6 induction in the center (Fig 7d and e, respectively, lower panels) are shown. In most of the periphery, SIRT6 was not induced with Tmx, and strong H3K9Ac staining was observed in general; however, H3K9Ac staining was less intense in the peripheral cells with sporadic SIRT6 induction (Fig 7d). In the center, SIRT6 was induced with Tmx in the RPE nuclei in large areas, and H3K9Ac staining was less intense in general; however, H3K9Ac levels were higher in the central cells with poor SIRT6 induction (Fig 7e). These results show that the system in our SIRT6 transgenic mice was functioning as expected.

Lastly, the most critical question was whether SIRT6 remains in the nucleus after $NaIO_3$ injection in our transgenic mice. We induced SIRT6 by giving Tmx for 3 days, injected $NaIO_3$ (20 mg/kg BW) on day 4, and analyzed SIRT6 1 h later using immunofluorescence of RPE flat-mounts. In Wt mice, although SIRT6 was detected exclusively in the nucleus in control without $NaIO_3$, nuclear SIRT6 quickly decreased and became barely detectable by 1 h after $NaIO_3$ injection (Fig 7f, left 2 panels). In SIRT6 transgenic mice (S4 line), although $NaIO_3$ caused a quick reduction of nuclear SIRT6 with the vehicle, SIRT6 was still intensely detected in the nucleus with Tmx even after $NaIO_3$ injection, especially in the center (Fig 7f, right 2 panels). We confirmed these observations with higher magnification images (Fig 7g, left panels) and quantification of SIRT6 signal intensity in the images (Fig S4, left panel). Concomitantly, H3K9Ac levels decreased in the S4 mice with Tmx (Fig 7g, right panels, and Fig S4, right panel). All images were taken using the same setting with low gain to avoid signal saturation (Fig 7f and g). These results show that SIRT6 was induced by Tmx predominantly in the center and that the induced SIRT6 largely remained in the nucleus even under oxidative stress.

## SIRT6 overexpression in the nucleus protects the RPE from oxidative stress in mice

The aim of generating the SIRT6 transgenic mice was to analyze the effect of SIRT6 overexpression in the nucleus on $NaIO_3$-induced RPE damage. In the prevention scheme, SIRT6 was induced in SIRT6 transgenic mice (S1 and S4 lines) with Tmx for 5 days (days 1–5), then $NaIO_3$ was injected on day 6, followed by additional Tmx every 2 days, and RPE morphology was analyzed on day 13 (Fig S5). We used a low dose of $NaIO_3$ (15 mg/kg BW) to produce milder oxidative stress so that we could more easily detect the protective effects of SIRT6. Representative images of RPE flat-mounts with ZO-1 immunofluorescence from the vehicle group showed the typical three distinct regions of RPE damage, periphery (normal RPE), transitional zone (elongated RPE), and center (damaged RPE), (A, B, and C, respectively, in Fig 8a, top panels). Representative images from the mice with successful protection in the Tmx group showed the normal cobblestone-like appearance in the entire RPE flat-mounts (A, B, and C in Fig 8a, bottom panels). Using our quantification method described earlier (Fig 1e), each area of the three regions was measured on RPE flat-mounts, and the proportion of each region to the entire RPE was calculated. In both S1 and S4 lines, Tmx treatment significantly preserved mouse RPE, with a larger area of normal-shaped RPE and a smaller area of damaged RPE compared with the vehicle (Fig 8b). Particularly in the S4 line, the RPE was completely preserved, at least morphologically, in 8 of 12 mice with Tmx ($P < 0.0001$). The transitional zone was consistently at ~10% when it existed, regardless of the vehicle or Tmx. In the treatment scheme, $NaIO_3$ was injected first, followed by Tmx injections, and RPE damage was quantified 7 days after $NaIO_3$ injection using the same quantification method (Fig S5). In either S1 or S4 line, there was no difference between the vehicle and Tmx groups in the proportion of any area (Fig S6). These results show that SIRT6 overexpression in the nucleus protected mouse RPE from oxidative stress but did not reverse RPE damage once it occurred in our experimental conditions.

To gain mechanistic insights into the protective effects, we analyzed the expression of selected genes in SIRT6 transgenic mice (S4 line). SIRT6 was induced with Tmx for 3 days, followed by $NaIO_3$ injection on day 4, and mRNA levels in the RPE were analyzed 6 h later using RT–qPCR. We used this early time point because mRNA levels of the genes selected were increased or decreased quickly by 6 h in our preliminary studies. In these experiments, we used a high dose of $NaIO_3$ (60 mg/kg BW) to strongly up-regulate NF-κB targets and thereby make it easier to detect their repression. We first analyzed the mRNA levels of RPE markers to check the integrity of

---

normalized by those of β-actin (SIRT6/β-actin), and relative SIRT6 levels were calculated as the ratio of SIRT6/β-actin in each sample to that in wild-type RPE (Wt, w). Results are presented as mean ± SEM with individual data points from the two gels (bottom panel). Statistical significance is shown by \*$P < 0.05$ (one-way ANOVA), without marking ns (not significant) in the graph. **(d, e)** Immunofluorescence of mouse RPE flat-mounts. SIRT6 transgenic mice (S4 line) were injected i.p. with Tmx or vehicle for 3 days (days 1–3), and SIRT6 (green) and acetylated H3K9 (H3K9Ac, red) were analyzed along with DAPI (blue) on day 4. All images were taken using the same microscopic settings. Representative images are shown from the periphery (d) and center (e) (upper panels). An area with sporadic SIRT6 induction in the periphery (d) and a patchy area with poor SIRT6 induction in the center (e) are also shown (lower panels). **(f)** Immunofluorescence of mouse RPE flat-mounts for SIRT6 distribution. SIRT6 transgenic mice (S4 line) were injected i.p. with Tmx or vehicle for 3 days (days 1–3), $NaIO_3$ (20 mg/kg BW) was injected on day 4, and SIRT6 (red) was stained 1 h later. In parallel, Wt mice were injected with $NaIO_3$ and analyzed for SIRT6 in the same manner. All images were taken using the same low exposure settings (for SIRT6 [Alexa 546]: laser transmission, 80%; gain, 900/1,250; digital offset, –0.04) to avoid signal saturation. Representative images are shown. **(g)** Higher magnification images of the RPE flat-mounts in (f). H3K9Ac (green) was also analyzed on RPE flat-mounts in the same manner as described in (f). The signal intensity of SIRT6 and H3K9Ac in the DAPI-stained nuclei in the images was quantified and included in Fig S4. Scale bars in the images: (a) 1,000 μm; (b, f) 500 μm; (d, e) 20 μm; (g) 50 μm.

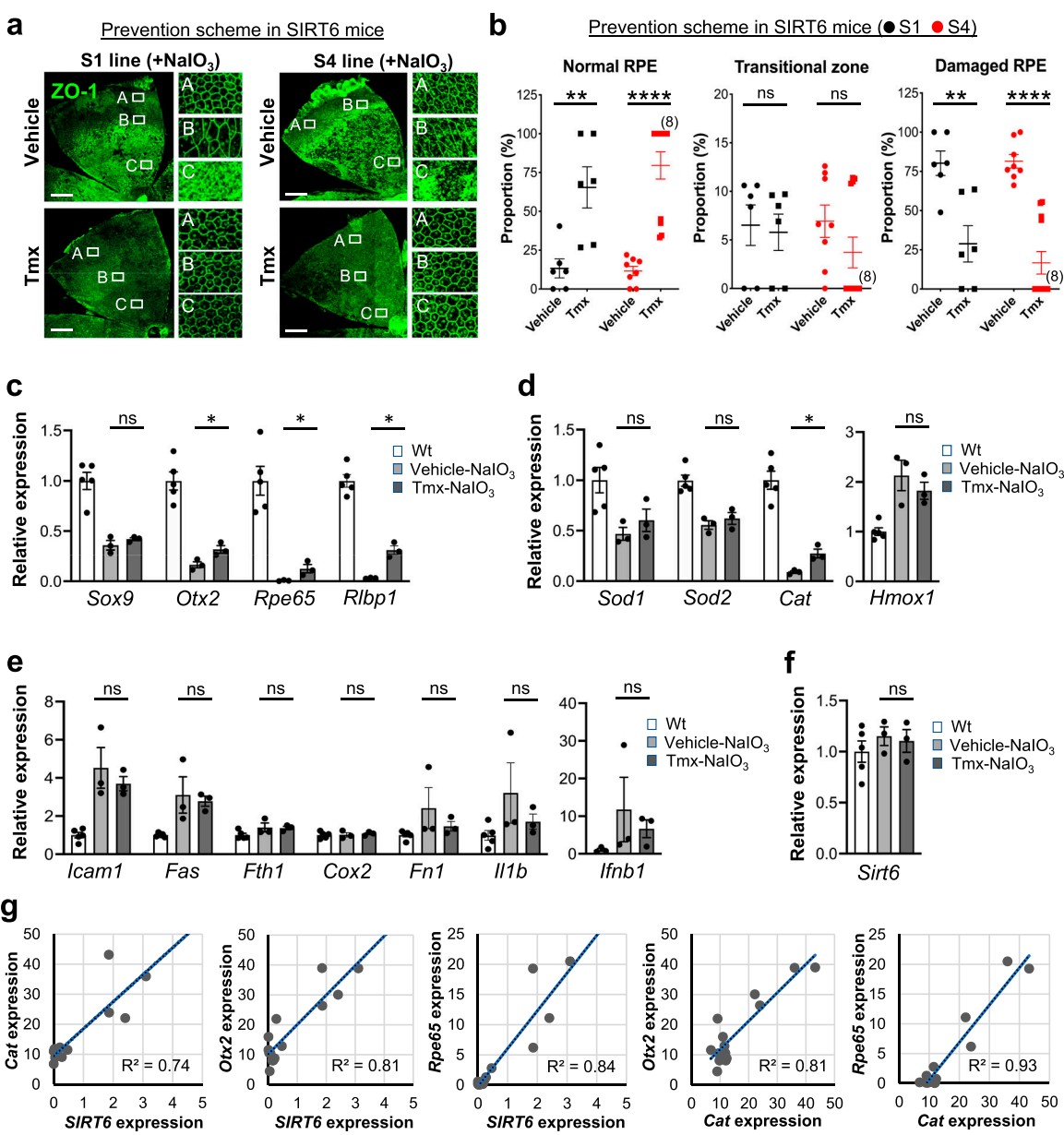

**Figure 8. SIRT6 overexpression protects mouse RPE from oxidative stress and partially preserves catalase expression.**
**(a, b)** The effects of SIRT6 overexpression on NaIO$_3$-induced RPE damage in the prevention scheme. SIRT6 transgenic mice (S1 and S4 lines) were injected i.p. with Tmx (0.5 mg/mouse/day) or vehicle (10% ethanol in sunflower oil) for 5 days (days 1–5), NaIO$_3$ (15 mg/kg BW) was injected on day 6, additional Tmx was given every 2 days (days 7, 9, and 11), and RPE damage was analyzed on day 13 (Fig S5). **(a)** Representative images of RPE flat-mounts with ZO-1 immunofluorescence from typical cases in the vehicle group and successfully protected cases in the Tmx group. Higher magnification images of the boxed areas A, B, and C are shown on the right side. Scale bars in the images: 500 μm. **(b)** Quantification of RPE damage displaying the three distinct regions: periphery (normal RPE), transitional zone, and center (damaged RPE). Using our quantification method (Fig 1e), each region was measured on RPE flat-mounts with ZO-1 immunofluorescence, and the proportion of each region to the entire RPE was calculated. Results are presented as mean ± SEM with individual data points (n = 6 mice for each S1 group; n = 8–12 mice for each S4 group). In the S4 line, the RPE was completely preserved morphologically in 8 of 12 mice with Tmx treatment. **(c, d, e, f)** The effects of SIRT6 overexpression on NaIO$_3$-induced gene expression changes. SIRT6 transgenic mice (S4 line) were injected with Tmx for 3 days (days 1–3), NaIO$_3$ (60 mg/kg BW) was injected on day 4, and total RNA was prepared from mouse RPE without the choroid 6 h later. The mRNA levels were analyzed by RT–qPCR for (c) RPE markers, (d) antioxidant genes, (e) NF-κB target genes, and (f) endogenous mouse *Sirt6*. The mRNA levels were calculated using the 2$^{-\Delta\Delta Ct}$ method with a geometric mean of three reference genes, *Gapdh*, *Hprt1*, and *Rplp0*, for normalization. Relative expression is presented as the ratio to the mRNA level in control Wt mice. Results are presented as mean ± SEM with individual data points (n = 3–5 mice for each group). **(g)** A linear regression analysis of mRNA levels. The relationship of mRNA levels between the transgene human *SIRT6* and endogenous mouse genes was analyzed by linear regression. *SIRT6* was discriminated from mouse *Sirt6* by human gene-specific primers for RT–qPCR, and relative expression (arbitrary unit) was calculated as the ratio to the mouse *Sirt6* mRNA level in Wt mice. The relative expression of *SIRT6* and mouse genes in the S4 mice was multiplied by 100 for analysis. Statistical significance throughout the figure: ns (not significant), \*P < 0.05, \*\*P < 0.01, and \*\*\*\*P < 0.0001 (t test).

RPE. The expression of *Otx2*, *Rpe65*, and *Rlbp1* was significantly preserved with Tmx compared with the vehicle (Fig 8c). We next analyzed the mRNA levels of antioxidant genes. SIRT6 over-expression with Tmx partially preserved *Cat* (catalase) expression; however, the mRNA levels of *Sod1*, *Sod2*, and *Hmox1* were similar with the vehicle and Tmx (Fig 8d). Because SIRT6 was reported to repress NF-κB targets by deacetylating H3K9Ac at the promoters (Kawahara et al, 2009, 2011), and NF-κB is one of the key regulators of stress response, repression of NF-κB targets could be a possible mechanism of the observed effects of SIRT6. Therefore, we analyzed the mRNA levels of NF-κB targets. SIRT6 overexpression induced by Tmx showed a mild trend of suppressing the up-regulation of NF-κB targets, but such effects were not significant (Fig 8e). SIRT6 in-duction with Tmx did not change the level of endogenous mouse *Sirt6* (Fig 8f). These results show that SIRT6 overexpression mod-estly but significantly preserved, although far from complete, the expression of RPE markers and catalase compared with control.

Lastly, we analyzed the relationship of mRNA levels between the transgene *SIRT6* and antioxidant *Cat* or RPE markers *Otx2* and *Rpe65* and between *Cat* and these RPE markers using linear regression (Fig 8g). Because human *SIRT6* was used to make the transgenic mouse line, we could discriminate *SIRT6* from endogenous mouse *Sirt6* using *SIRT6*-specific primers for RT–qPCR. Relative expression in the S4 line with Tmx or vehicle was calculated as the ratio to the level in control WT mice and multiplied by 100 for each gene. The ex-pression of *SIRT6* was positively correlated with that of *Cat*, *Otx2*, and *Rpe65*, and the expression of *Otx2* and *Rpe65* was well cor-related with that of *Cat* (Fig 8g). These results suggest that the protective effects of SIRT6 overexpression against oxidative stress may rely on preserved catalase.

## Discussion

The RPE has high metabolic activities and constantly faces oxidative stress because of daily phagocytosis of photoreceptor outer seg-ments containing photooxidized molecules and its close proximity to high blood flow in the choriocapillaris (Strauss, 2005). As an oxidative stress model in vivo, a high dose of NaIO₃ that is primarily toxic to the RPE has long been used to study retinal degeneration following RPE death, and thus, characterization of RPE damage itself is still limited. It has been reported that SIRT6 is protective against oxidative stress (Pan et al, 2016; Wang et al, 2016; Ka et al, 2017; Kim et al, 2019; Yu et al, 2019); however, SIRT6's role in the RPE is still largely unknown. In this study, we aimed to address these two understudied questions by characterizing NaIO₃-caused RPE damage in mice and testing the functional role of SIRT6 in oxidative stress-induced RPE damage.

### Topological differences in the susceptibility of mouse RPE to oxidative stress

With a low dose of NaIO₃, we observed that RPE changes could be morphologically divided into three regions on RPE flat-mounts as previously reported: periphery (normal RPE), transitional zone (elongated cells), and center (damaged or lost RPE) (Xia et al, 2011).

We suspected that each region represented the different degrees of RPE damage, that is, mild (periphery), intermediate (transitional zone), and severe (center), and that the ratio of these regions could indicate the severity of overall RPE damage. Therefore, we devel-oped a method to quantify each region by image analyses. For the observed topological difference that central RPE near the optic nerve head was more susceptible to NaIO₃ than peripheral RPE, we considered two mechanisms: (1) choroidal vascular circulation that flows from center to periphery and therefore creates a NaIO₃ gradient and (2) differences in intrinsic RPE cell properties. To eliminate the vascular influence, we made ex vivo mouse RPE and found that RPE cells in the center were still more susceptible to oxidative stress, supporting cell intrinsic differences. However, the two mechanisms are not mutually exclusive, and we speculate that both are likely involved. Although mice do not have a macula, our results that central RPE is more susceptible to oxidative stress may have important implications for using mice as a model. Interest-ingly, the central region of mouse retina has a higher photoreceptor cell density and a larger RPE cell size than the periphery, resulting in a higher phagocytic load per RPE cell (Volland et al, 2015). In addition, the phagocytic load per RPE cell is greater in the mouse central retina than in the human macula, suggesting that the or-ganizational characteristics may make the mouse central retina a sensitive model for at least the peripheral part of human macula (Volland et al, 2015). As for retinal degeneration by NaIO₃, it was reported that whereas a high dose of NaIO₃ (40 mg/kg BW) caused degeneration of the whole retina, a low dose (20 mg/kg BW) induced degeneration in the central retina with no or mild damage in the periphery (Machalinska et al, 2010). This report focused on retinal degeneration, but it might have reflected RPE damage. The topological differences in the RPE susceptibility to NaIO₃ have been reported in mice (Xia et al, 2011; Ma et al, 2020; Upadhyay et al, 2020; Wolk et al, 2020; Zhang et al, 2021); however, our study is the first to show that such differences can also be because of intrinsic RPE cell properties.

Of great interest and relevance, the topological difference of human RPE was recently described in detail (Ortolan et al, 2022). Using an artificial intelligence–based approach, a single–cell–resolution morphometric map of entire human RPE was generated from fluorescently labeled RPE flat-mounts, and cell area analyses revealed five different RPE subpopulations locating in concentric circles. Importantly, the authors found that different retinal de-generative diseases affected different RPE subpopulations (Ortolan et al, 2022). Thus, RPE heterogeneity will be an important subject regarding the susceptibility to retinal degeneration in future studies.

### EMT as part of RPE damage caused by oxidative stress

To characterize the three regions of NaIO₃-caused RPE damage in mice, we analyzed molecular changes. In the periphery with normal-shaped RPE, although TUNEL assays suggested that the RPE maintained enough functions to support the survival of photore-ceptors, SNAI1 was already up-regulated. In the transitional zone with elongated RPE, SNAI1 and ZEB1 were up-regulated with de-creased RPE markers, suggesting that RPE might be undergoing EMT. In the retina facing this zone, TUNEL staining was weaker but clearly positive, showing that photoreceptors began to die in this region. In the center with severely damaged RPE, extensive TUNEL

staining indicated massive photoreceptor deaths. Analyses of RNAs from the entire RPE showed the large decrease of RPE markers such as visual cycle genes, suggesting that even surviving RPE cells might not be fully functional.

EMT is a process in which epithelial cells lose epithelial phenotypes and become fibroblastic cells with increased mesenchymal markers (Lamouille et al, 2014; Dongre & Weinberg, 2019). EMT can exhibit a wide range of changes from partial to complete EMT (Dongre & Weinberg, 2019), and our elongated RPE cells in the transitional zone seem to represent partial EMT. In a mouse model of kidney fibrosis, injuries induce partial EMT of renal tubular epithelial cells that remain on the basement membrane but secrete pro-inflammatory cytokines and chemokines, which promote differentiation of interstitial cells to myofibroblasts and sustain inflammation (Grande et al, 2015). Such outcomes are attenuated by the deletion of *Snai1* or *Twist1* (Grande et al, 2015; Lovisa et al, 2015). Based on these results in mouse kidney, it is conceivable that NaIO$_3$ caused partial EMT of RPE cells that remained in the RPE layer but affected other RPE and non-RPE cells through secreted factors in our mice. Regarding EMT of RPE cells, extensive EMT was reported in RPE-specific *Pten* conditional knockout (cko) mice, in which RPE cells migrated out of the retina (Kim et al, 2008). It is unclear whether such extensive EMT also occurred in our mice. Because PTEN interacts with proteins in adherens junctions, the phenotype of *Pten* cko mice underscores the importance of cell adhesion for the integrity of RPE. Importantly, a loss of PTEN's interaction with junctional proteins was also observed in mice with NaIO$_3$ (Kim et al, 2008). We previously reported that P-cadherin and $\beta$-catenin in adherens junctions were dislocated from the cell membrane to the cytoplasm after NaIO$_3$ injection, followed by translocation of $\beta$-catenin to the nucleus and up-regulation of its target *Snai1* (Yang et al, 2018). EMT or dedifferentiation has been recognized as RPE response to stresses. In RPE-selective cko mice for *Tfam*, the gene encoding mitochondrial transcription factor A, oxidative phosphorylation in the mitochondria was disrupted, and RPE cells gradually dedifferentiated to hypertrophic cells with reduced RPE markers (Zhao et al, 2011). RPE dedifferentiation was also seen after NaIO$_3$ injection in WT mice, suggesting that it was a common response to metabolic and oxidative stresses (Zhao et al, 2011). Consistent with these findings, it was recently reported that impaired mitophagy and mitochondrial dysfunction because of the deficiency of PINK1, a mitochondrial protein kinase initiating mitophagy, triggered RPE EMT through retrograde mitochondrial–nuclear signaling that led to the up-regulation of SNAI1 and ZEB1 with an EMT-like transcriptome (Datta et al, 2023). When NRF2 was also deleted, EMT morphology was normalized but RPE cells died (Datta et al, 2023).

Human donor eyes provide clues about the role of RPE EMT in eye diseases. In the RPE/choroid with geographic atrophy (GA), an advanced form of dry AMD, three regions (non-atrophic, border, and atrophic) were observed, with hypertrophic RPE at the border (McLeod et al, 2009). In wet AMD, RPE hypertrophy was also seen at the edge of choroidal neovascularization (CNV) (McLeod et al, 2009). Whether RPE hypertrophy in these GA and CNV eyes represents EMT is unclear without molecular signatures, but it is remarkably similar to the morphological changes observed in the *Tfam* cko mice described above and in our mice with NaIO$_3$. Importantly, the RPE/

choroid of human dry AMD eyes had increased SNAI1 and vimentin with decreased E-cadherin compared with controls (Ghosh et al, 2018). The above described study also reported that the levels of PINK1 and NRF2 decreased in dysmorphic perifoveal RPE of early AMD eyes, suggesting that the two cytoprotective mechanisms, PINK1-mediated mitophagy and NRF2-dependent antioxidant defense, are weakened in AMD and likely contribute to the observed RPE heterogeneity (Wang et al, 2014; Datta et al, 2023). Clinically, RPE EMT plays a key role in PVR, and fibrous epiretinal membranes (ERMs) in PVR are produced by RPE cells undergoing EMT (Tamiya & Kaplan, 2016; Chaudhary et al, 2020). In the vitreous of PVR patients, TGF-$\beta$ increased and correlated with disease severity (Kita et al, 2008), and patient-derived ERMs showed activation of TGF-$\beta$ and TNF-$\alpha$ signaling (Asato et al, 2013). When human stem cell-derived RPE cells were co-treated with TGF-$\beta$ and TNF-$\alpha$, they produced fibroblastic contractile membranes resembling ERMs (Boles et al, 2020). Thus, the role of RPE EMT is well established in PVR. In the case of AMD, although further studies are needed to clarify the exact role of RPE EMT, relevant data are being accumulated.

## Protective effects of SIRT6 overexpression in the nucleus against oxidative stress

SIRT6 plays a role in a variety of biological processes through its enzymatic activities (Chang et al, 2020). SIRT6 regulates glucose homeostasis by repressing glycolytic genes as a corepressor of HIF1$\alpha$ (Zhong et al, 2010), suppresses NF-$\kappa$B targets as a histone deacetylase (Kawahara et al, 2009), and stimulates DNA repair through PARP1 activation as a mono-ADP-ribosylase (Mao et al, 2011; Xu et al, 2015). In addition, SIRT6 acts as a coactivator of NRF2 in regulating antioxidant genes in human mesenchymal stem cells under oxidative stress (Pan et al, 2016). Interestingly, lamin A is an endogenous activator of SIRT6 (Ghosh et al, 2015) and NRF2 interacts with lamin A at the nuclear periphery (Kubben et al, 2016), suggesting that lamin A, SIRT6, and NRF2 form functional networks (Gorbunova et al, 2016). All of these important functions happen in the nucleus. However, it was also reported that following heat shock, SIRT6 rapidly translocated to the cytoplasm, interacted with G3BP1, and regulated the assembly and disassembly of SGs in mouse embryonic fibroblasts (Jedrusik-Bode et al, 2013; Simeoni et al, 2013). SG is a cytoplasmic membrane-less organelle, which is assembled in response to diverse stresses by liquid–liquid phase separation (LLPS) and disassembled during stress recovery, and contains RNA–protein complexes with translationally stalled mRNAs (Protter & Parker, 2016; Wheeler et al, 2016; Wolozin and Ivanov, 2019). Proteins undergoing LLPS contain low-complexity sequences or intrinsically disordered regions that can form the weak multivalent macromolecular interactions important for LLPS (Banani et al, 2017). Importantly, SIRT6 is predicted to contain an intrinsically disordered region at the C-terminus (Miteva & Cristea, 2014; Klein & Denu, 2020), suggesting that SIRT6 could physically participate in LLPS to form SGs in the cytoplasm. *Sirt6*-deficient cells showed the disruption of SG assembly with increased cell death, suggesting SIRT6's role in regulating SGs for cell survival (Jedrusik-Bode et al, 2013; Simeoni et al, 2013). However, because these studies focused on SG assembly in the cytoplasm, it is unclear whether increased cell death might also result from a lack of SIRT6

in the nucleus. This SIRT6's cytoplasmic function has not been further studied in detail or in other cell types. In the present study, we observed that SIRT6 quickly moved from the nucleus to the cytoplasm in mouse RPE after both oxidative stress in vivo and physical stress ex vivo and that SIRT6 colocalized with G3BP1 in the cytoplasm but did not return to the nucleus after G3BP1 aggregates disappeared, resulting in SIRT6 depletion in the nucleus. Thus, although regulating SG assembly is important, SIRT6 depletion in the nucleus would hinder its critical functions. To overcome this problem, we generated transgenic mice with inducible SIRT6 overexpression in the nucleus using $ER^{T2}$ combined with the RPE-preferential *BEST1* promoter. In these mice, SIRT6 was induced by Tmx, stayed in the nucleus even under oxidative stress, and indeed protected the RPE in our prevention scheme. Because induced SIRT6 did not reverse RPE damage in our treatment scheme, this failure may be because of a time lag from Tmx injection to SIRT6 induction, the likelihood of SIRT6 acting at the early stage of oxidative damage as discussed below, and/or irreversible changes occurring quickly. Thus, SIRT6 seems to have the potential for protecting against, rather than treatment for, oxidative stress-caused RPE damage.

SIRT6's roles in the retina have been reported, including the essential role in adult retinal function (Silberman et al, 2014), the protection of retinal ganglion cells from $H_2O_2$-induced oxidative damage through NRF2 signaling (Yu et al, 2019), and the effect on energy metabolism by inhibiting glycolytic flux in photoreceptors (Zhang et al, 2016). In the RPE, however, studies of SIRT6 are still scarce, except two reports describing activation of autophagy by SIRT6 with conflicting effects on inflammation (Feng et al, 2018; Liu & Liu, 2020). SIRT6's protective effects against oxidative stress have been reported in other tissues (Pan et al, 2016; Wang et al, 2016; Ka et al, 2017; Kim et al, 2019), but our study is the first to show such effects in the RPE regardless of in vitro or in vivo.

### Protective effects of SIRT6 overexpression against oxidative stress is possibly mediated by preservation of catalase expression

We tried to gain insights into the mechanisms of SIRT6's effects by analyzing gene expression changes. It is known that SIRT6 interacts with p65 of NF-κB, thereby is recruited to the promoter of NF-κB target genes, and suppresses their expression by H3K9 deacetylation (Kawahara et al, 2009, 2011). Therefore, we first suspected that suppression of NF-κB targets might be the mechanism by which SIRT6 protected RPE cells from oxidative stress. NF-κB acts at the center of cellular response to various external and internal stresses including oxidative stress (Oeckinghaus & Ghosh, 2009; Hayden & Ghosh, 2012; Sivandzade et al, 2019). In addition, EMT-TFs, such as *Snai1*, *Twist1*, *Zeb1*, and *Zeb2*, and mesenchymal genes, such as *Vim* and *Mmp9*, are direct targets of NF-κB (Huber et al, 2004; Tobar et al, 2010; Cichon & Radisky, 2014; Lamouille et al, 2014; Liu et al, 2015; Dongre & Weinberg, 2019), suggesting that NF-κB likely plays a role in multiple aspects of our NaIO$_3$ model. Supporting such roles of NF-κB, we recently reported that IKKβ inhibitor BAY 651942, an upstream inhibitor of NF-κB signaling, protected mouse RPE from NaIO$_3$ in an experimental condition similar to our prevention scheme (Yang et al, 2021). Although this IKKβ inhibitor suppressed

the up-regulation of NF-κB targets induced by NaIO$_3$ as expected, SIRT6 overexpression showed no significant suppression of NF-κB targets in the present study. Instead, SIRT6 overexpression partially preserved the expression of *Cat* (catalase), an antioxidant enzyme that degrades $H_2O_2$ to $H_2O$ and $O_2$ (Ho et al, 2004; Goyal & Basak, 2010). These results suggest that the effect of SIRT6 overexpression is unlikely mediated by the suppression of NF-κB targets in our mice. Grimley et al. also reported that overexpression of SIRT6 had little effect on NF-κB target gene expression (Grimley et al, 2012).

In the present study, the mRNA levels were correlated between *SIRT6* and *Cat* and between *Cat* and RPE markers *Otx2* and *Rpe65* in our transgenic mice. In addition, we recently found that the RPE was more susceptible to NaIO$_3$ in female mice than in male mice, and that the mRNA level of RPE markers, particularly *Otx2* and *Rlbp1*, was well correlated with that of *Cat* (Yang et al, 2021). Of great interest, it was reported that catalase transduced in mouse RPE using an adenovirus vector protected the neighboring photoreceptors from oxidative stress in a light damage model (Rex et al, 2004). Therefore, we speculate that preservation of catalase is one of the mechanisms by which SIRT6 protected the RPE in our mice. Interestingly, general SIRT6 over-expression and mitochondrial catalase overexpression both extend the mouse lifespan (Schriner et al, 2005; Kanfi et al, 2012). Although NRF2 is a key regulator of many antioxidant enzymes through binding to antioxidant response elements (AREs) in the gene promoter (Ma, 2013; Tonelli et al, 2018), it is still controversial whether NRF2 regulates *Cat* because the regulatory region of *Cat* has no AREs (Glorieux et al, 2015). Of note, myocardial damage in ischemia/reperfusion was aggravated in *Sirt6*$^{+/-}$ mice, but this effect was reversed with an adenovirus carrying *SIRT6* (Wang et al, 2016). Mechanistically, SIRT6 activated FOXO3 in an AMPK-dependent manner and thereby up-regulated FOXO-dependent antioxidant genes including *Cat*. This SIRT6–AMPK–FOXO3 axis may also be involved in our NaIO$_3$ model.

### Summary

RPE damage caused by NaIO$_3$-induced oxidative stress in mice was divided into three regions, periphery, transitional zone, and center, with distinct RPE morphologies. RPE cells in the transitional zone showed characteristics of partial EMT. Central RPE was more susceptible to oxidative and physical stresses than peripheral RPE. In response to stress, SIRT6 translocated from the nucleus to the cytoplasm, resulting in nuclear SIRT6 depletion. SIRT6 over-expression in the nucleus significantly protected the RPE from oxidative stress in transgenic mice. This protective effect was correlated with catalase expression. Our results warrant further studies of SIRT6 as a potential target for protecting RPE cells from oxidative stress-induced damage.

# Materials and Methods

### Mice

All mice were treated in accordance with the Federal Guide for the Care and Use of Laboratory Animals and the guidelines of the Johns Hopkins University Institutional Animal Care and Use Committee

(IACUC). For all animal experiments involving Wt mice, we used 8–12 wk old male C57BL/6J mice (Jackson Laboratory) unless specified otherwise. All transgenic mice were maintained on the C57BL/6J background.

## Generation of transgenic mice

Transgenic mice carrying *BEST1* −585/+38 promoter–*lacZ* were generated and analyzed by X-gal staining in our previous studies (Esumi et al, 2004) (Fig 7a, bottom). For the present study, we generated a new transgenic mouse model with inducible SIRT6 overexpression in the RPE. In this transgenic model, the 4-hydroxytamoxifen (4-OHT)-responsive mutant estrogen receptor ER$^{T2}$ (a gift from Pierre Chambon, France) (Feil et al, 1997; Indra et al, 1999) was combined with the human *BEST1* −585/+38 promoter that drives the expression in the RPE (from our work) (Esumi et al, 2004, 2007). Inducible SIRT6 overexpression is achieved by in-frame fusion of human SIRT6 coding sequence with the ER$^{T2}$ (Fig 7a, top). In this mouse model, the fusion protein SIRT6–ER$^{T2}$ is normally sequestered in the cytoplasm and inactive; upon addition of 4-OHT, it translocates to the nucleus and exerts its effects. Microinjection of the transgenic construct was performed at the transgenic core facility of Johns Hopkins University School of Medicine. Out of 79 mouse pups, seven were positive for *BEST1–SIRT6–ER$^{T2}$* by genotyping with primers producing a DNA fragment encompassing the junction of *ER$^{T2}$* and placF vector (Table S1). The expression of the transgene was confirmed by RT–PCR that showed the expected size.

## Tmx administration

We tested the effects of SIRT6 induction on RPE morphologies in two experimental schemes, prevention and treatment (Fig S5). For the prevention scheme, adult SIRT6–ER$^{T2}$ transgenic mice (3–6 mo old) were injected i.p. with Tmx (0.5 mg/mouse/day) (Indra et al, 1999) or vehicle (10% ethanol in sunflower oil) for five consecutive days (days 1–5). NaIO$_3$ was injected on day 6, additional Tmx was given every other day (days 7, 9, and 11), and the mice were euthanized for analyses on day 13 (7 days after NaIO$_3$ injection). For the treatment scheme, NaIO$_3$ was injected first on day 1, followed by Tmx injections i.p. twice on day 1 (immediately after NaIO$_3$ and 4 h later) and once a day on additional 5 days (days 2–5 and 7). The mice were euthanized for analyses on day 8 (7 days after NaIO$_3$ injection). For analyses of the effects of SIRT6 induction on gene expression, SIRT6–ER$^{T2}$ transgenic mice were injected i.p. with Tmx or vehicle for 3 days (days 1–3), NaIO$_3$ was injected on day 4, and the mice were euthanized for analyses 6 h later on the same day.

## NaIO$_3$ injection

To induce oxidative stress in mice, NaIO$_3$ (S4077; MilliporeSigma) in 200 µl of PBS was injected through the tail vein. For analyses of RPE damage (RPE flat-mounts), retinal degeneration (TUNEL assays), mRNA expression (RT–qPCR), and protein expression (Western blots), NaIO$_3$ was used at 15 mg/kg BW. For analyzing SIRT6 translocation from the nucleus to the cytoplasm, NaIO$_3$ was used at 20 mg/kg BW to give a stronger oxidative stress. For analyses of the

early effects of SIRT6 induction on gene expression, NaIO$_3$ was injected at 60 mg/kg BW to strongly up-regulate NF-κB target genes.

## Immunofluorescence of retinal sections

Mouse eyes were fixed in 4% PFA in 0.1 M phosphate buffer at 4°C for 30 min, a hole was created in the center of the cornea, and the eyes with a hole were fixed in the same 4% PFA solution at 4°C for additional 30 min. The fixed eyes were cryoprotected through an increasing gradient of sucrose in 0.1 M phosphate buffer (6.25% sucrose, on ice for 45 min; 12.5%, on ice for 45 min; and 25%, at 4°C overnight) followed by embedding in OCT Tissue-Tek (4583; Sakura Finetek USA) for cryostat. Immunofluorescence of frozen eye sections was performed as described previously (Masuda et al, 2014). Primary antibodies used are listed in Table S2. Secondary antibodies were anti-mouse, anti-rabbit or anti-goat IgG conjugated with Alexa Fluor 488, 549 or 647 (1:500; Invitrogen, Thermo Fisher Scientific). To observe signals clearer in the RPE, melanin pigment was bleached following the published protocol (Bhutto et al, 2004). Nuclei were stained with 4′,6-diamidino-2′-phenylindole dihydrochloride (DAPI, 10236276001; Roche) at room temperature for 10 min. The sections were mounted in a Fluorescent Mounting Medium (S3023; Dako), and images were acquired using an LSM 510 inverted laser scanning confocal microscope (Carl Zeiss).

## RPE/choroid flat-mounts

RPE/choroid flat-mounts (called RPE flat-mounts) were made and stained as described previously (Yang et al, 2018). Briefly, mouse eyes were dissected at the equator, the cornea and lens were removed, and the neural retina was peeled off. The eyecups consisting of the RPE, choroid and sclera were fixed in 4% PFA in 0.1 M phosphate buffer at room temperature for 10 min and transferred into PBS. The eyecups were partially cut into quarters by four radial cuts from the periphery toward the optic disc and blocked in TBS with 0.25% Triton X-100, 10% normal horse serum (Z0610; Vector Laboratories), and 1% bovine serum albumin (BSA, A9647; MilliporeSigma) at room temperature for 1 h. The RPE flat-mounts were then incubated with a desired primary antibody (Table S2) at 4°C overnight. After washing with TBS at room temperature for 5 min three times, appropriate secondary antibodies described above for eye sections were added. Nuclei were stained with DAPI (10236276001; Roche) at room temperature for 10 min. The flat-mounts were washed with TBS for 5 min three times and mounted in the Fluorescent Mounting Medium (S3023; Dako), with the RPE side facing up. Images of the RPE flat-mounts were acquired using the LSM 510 confocal microscope. For staining F-actin (filamentous actin), the RPE flat-mounts were incubated with Cyto-Painter Phalloidin-Fluor 555 Reagent (1:1,000; ab176756; Abcam) in PBS with 1% BSA at room temperature for 30 min.

## Quantification of mouse RPE damage

To quantify RPE damage by NaIO$_3$ in mice, we developed a morphometric method based on RPE flat-mounts with immunofluorescence for ZO-1 that is located at the RPE cell border and therefore outlines the shape of RPE cells. We divided RPE damage

into three regions with distinct RPE morphologies: normal cobblestone-like RPE (periphery), elongated RPE cells (transitional zone), and severely damaged RPE (center). The criteria for separating these three regions are described in detail in the Results section. RPE flat-mounts were made as described above, and images of the entire flat-mounts were acquired using the tiling function of LSM 510 confocal microscope. Then, each region was measured by the number of pixels using the ImageJ software (1.49v; National Institutes of Health [NIH]), and the proportion (%) of each region to the entire flat-mount was calculated.

## TUNEL staining

Apoptotic cell death of retinal photoreceptors was analyzed by the TUNEL (terminal deoxynucleotidyl transferase dUTP nick end labeling) method using In Situ Cell Death Detection Kit, TMR red (12156792910; Roche) according to the company's instructions. Frozen eye sections were treated in citric acid buffer on ice for 2 min and washed with PBS. Then, the TUNEL reagent was added onto the slides and incubated at 37°C for 30 min. Nuclei were stained with DAPI at room temperature for 10 min, and sections were mounted in the Fluorescent Mounting Medium (S3023; Dako). Images were taken using the LSM 510 confocal microscope.

## RNA extraction from mouse RPE

To analyze gene expression in mouse RPE more accurately, we previously modified the RNA extraction method reported for only mouse RPE (Xin-Zhao Wang et al, 2012) to purify RNA from the RPE and choroid individually (Yang et al, 2018). The key modification was to employ two-step extraction, first with Trizol reagent (15596; Invitrogen, Thermo Fisher Scientific) and then with RNeasy Micro Kit (74004; QIAGEN). Trizol was used to lyse the choroid/sclera completely, and the RNeasy columns were used to remove pigments that inhibit enzymes. Briefly, we dissected mouse eyes and obtained the eyecup consisting of the RPE, choroid, and sclera. RPE cells were released by incubating the eyecup in RNAprotect cell reagent (76526; QIAGEN) for 10 min followed by gentle tapping of the tube. Then, we transferred the choroid/sclera eyecup to a new tube containing Trizol, collected the released RPE cells in the original tube by centrifugation, and added Trizol to the RPE pellets. After homogenizing the RPE pellets and the choroid/sclera eyecup using a pestle grinder, RNA was purified from each tissue by the two-step extraction.

## RT–qPCR

The mRNA expression of NF-κB targets, EMT markers, RPE markers, antioxidant genes, and other selected genes in mouse RPE was analyzed by RT–qPCR. Total RNA from mouse RPE was extracted using the two-step extraction method described above. RT–qPCR was performed as previously described (Esumi et al, 2009; Masuda & Esumi, 2010) with modifications. First-strand cDNA was synthesized from 200 ng of total RNA with random primers using SuperScript III reverse transcriptase (Invitrogen, Thermo Fisher Scientific), and real-time PCR was performed with gene-specific primers (Table S1) using C1000 Thermal Cycler (Bio-Rad). Relative gene expression values were calculated using the $2^{-\Delta\Delta Ct}$ method with a geometric mean of three reference genes, *Gapdh*, *Hprt1*, and *Rplp0* (or *Actb1*), for normalization. Each experimental group consisted of three to five biological replicates, and each sample was analyzed in triplicate in real-time PCR. The relationship of mRNA levels between the transgene *SIRT6* and endogenous mouse genes was analyzed using linear regression. The transgene, human *SIRT6*, was discriminated from endogenous mouse *Sirt6* using *SIRT6*-specific primers for RT–qPCR, and relative expression (arbitrary unit) was calculated as a ratio to the mouse *Sirt6* mRNA level in Wt mice. The relative expression of *SIRT6* and mouse genes was multiplied by 100 for analyses by linear regression.

## Measurement of absolute cDNA quantity

The absolute quantity of cDNA as an output of mRNA for *Cdh1*, *Cdh2*, and *Cdh3* in mouse RPE was measured after $NaIO_3$-induced oxidative stress as described previously (Yang et al, 2018). Briefly, cDNA fragments of *Cdh1*, *Cdh2*, and *Cdh3* for standard curves were generated by RT–PCR to contain the segment to be quantified later, purified using agarose gels, and the DNA concentration was measured using NanoDrop Spectrophotometer (Thermo Fisher Scientific). Because the expression of *Cdh1*, *Cdh2*, and *Cdh3* was barely detectable in the spleen, we diluted the gel-purified DNA fragments for standard curves into mouse spleen cDNA to obtain the molecular complexity similar to the samples. We used standard curves from 1 attomole (amole) to 0.1 zeptomole (zmole) that covered the threshold cycle (Ct) values of the samples. Sample cDNAs were synthesized from 200 ng of total RNA in 20 $\mu$l solution and diluted by 20-fold for RT–qPCR. Primers for RT–qPCR were designed to amplify cDNA fragments that encompass exon–exon borders. The final cDNA quantity was calculated for 200 ng of total RNA for each sample based on the process in which 1.5 $\mu$l of the 20-fold diluted cDNAs were used for real-time PCR. All primers are listed in Table S1.

## Western blot analysis

Mouse RPE protein lysates were prepared and analyzed by Western blotting as described previously (Yang et al, 2018). Briefly, mouse eyes were dissected to obtain the eyecup containing the RPE, choroid, and sclera. The eyecup was cut into four petals, incubated in RIPA lysis buffer containing protease inhibitors on ice for 45 min with occasional gentle tapping, and the insoluble fraction was removed by centrifugation. The supernatant was collected, the protein concentration was determined using a BCA protein assay kit (Pierce, Thermo Fisher Scientific), and 20 $\mu$g of proteins were used for SDS–PAGE and transferred onto a nitrocellulose membrane. The membrane was blocked in 5% nonfat dry milk in TBS with 0.1% Tween 20 (TBST) at room temperature for 30 min, incubated with a primary antibody (Table S2) in fresh 5% nonfat dry milk in TBST at 4°C overnight, washed in TBST, and incubated with an appropriate secondary antibody conjugated with HRP. The signals were detected with an ECL detection kit (RPN2232; GE Healthcare) using an ImageQuant LAS 4000 scanner (GE Healthcare). The intensity of each band was quantified using the ImageJ software (1.49v; NIH). The signal intensity of each protein was normalized by that of

$\beta$-actin (protein/$\beta$-actin), and relative protein levels were calculated as the ratio of protein/$\beta$-actin of samples to that of control.

## Ex vivo mouse RPE

To test whether there is an intrinsic difference in the susceptibility to stress between peripheral and central RPE, we developed ex vivo mouse RPE, which enables analyses without blood circulation that determines the local concentration of $NaIO_3$ at the RPE. We dissected mouse eyes and obtained the eyecup consisting of the RPE, choroid, and sclera. The eyecup was incubated in 1 ml of $CO_2$-independent medium (18045088; Thermo Fisher Scientific) in a microcentrifuge tube at 37°C for the desired duration (1–6 h). To induce oxidative stress, we added either $NaIO_3$ (7,500 $\mu g$/ml for 1 h), $H_2O_2$ (10 mM for 3 h; H1009; MilliporeSigma) or paraquat (2 mM for 3 h; 36541; MilliporeSigma) to the media. Then, the eyecup was fixed in 0.5 ml of 4% PFA in 0.1 M phosphate buffer at room temperature for 10 min, washed in PBS, and cut into quarters by four radial cuts from the periphery to make RPE flat-mounts. The ex vivo RPE flat-mounts were stained by ZO-1 immunofluorescence to determine the location of RPE damage.

## Statistical analysis

$t$ test (unpaired, two-tailed) and one-way ANOVA were used for statistical analyses with Prism 9 (GraphPad Software). A $P$-value less than 5% ($P < 0.05$) was regarded as statistical significance.

# Supplementary Information

# Acknowledgements

We would like to thank Dr. Pierre Chambon (Institute for Genetics and Cellular and Molecular Biology, France) for providing the $ER^{T2}$ construct. This work was supported by research grants from the BrightFocus Foundation (M2015220 to N Esumi), the US National Institutes of Health (R01EY016398 to N Esumi and core grant P30EY001765 to Wilmer Eye Institute), the Wilmer Pooled Professor Research Fund (PPF 2016 to N Esumi), and the Research to Prevent Blindness, Inc. (unrestricted funds to Wilmer Eye Institute).

## Author Contributions

X Yang: formal analysis, investigation, and methodology.
J-Y Chung: formal analysis, investigation, and methodology.
U Rai: investigation and methodology.
N Esumi: conceptualization, data curation, formal analysis, supervision, funding acquisition, validation, investigation, visualization, methodology, project administration, and writing—original draft, review, and editing.

## Conflict of Interest Statement

The authors declare that they have no conflict of interest.

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
