## [Reviewer comments · Life Science Alliance]

Life Science Alliance

SIRT6 overexpression in the nucleus protects mouse retinal pigment epithelium from oxidative stress

Xue Yang, Jin-Yong Chung, Usha Rai, and Noriko Esumi

DOI: <https://doi.org/10.26508/lsa.202201448>

Corresponding author(s): *Noriko Esumi, Johns Hopkins University*

Review Timeline:

Submission Date:	2022-03-14
Editorial Decision:	2022-05-06
Revision Received:	2023-02-28
Editorial Decision:	2023-03-29
Revision Received:	2023-04-05
Accepted:	2023-04-06

Scientific Editor: Novella Guidi

Transaction Report:

May 6, 2022

Re: Life Science Alliance manuscript #LSA-2022-01448-T

Dr. Noriko Esumi
Johns Hopkins University
Ophthalmology
400 North Broadway
Smith Building, Room 3041
Baltimore, Maryland 21231

Dear Dr. Esumi,

Thank you for submitting your manuscript entitled "SIRT6 overexpression in the nucleus protects mouse retinal pigment epithelium from oxidative stress" to Life Science Alliance. The manuscript was assessed by expert reviewers, whose comments are appended to this letter. We invite you to submit a revised manuscript addressing the Reviewer comments.

Thank you for this interesting contribution to Life Science Alliance. We are looking forward to receiving your revised manuscript.

Sincerely,

B. MANUSCRIPT ORGANIZATION AND FORMATTING:

Reviewer #1 (Comments to the Authors (Required)):

1. Using a sodium iodate (NaIO₃)-mediated method to induce oxidative stress-related RPE and retinal damage in mice, Yang, Chung et al. investigated the expression patterns of RPE and EMT factors at different time points post-NaIO₃ injection. The authors also examined regional changes in cellular SIRT6 localization and SIRT6 co-localization with G3BP1 post-NaIO₃ injection. In addition, transgenic mice were used to determine the protective effects of nuclear SIRT6 over-expression and impaired nuclear NF-κB signaling on RPE after NaIO₃-mediated damage. While oxidative stress-induced RPE damage is an active and interesting area of research, this study is lacking quantitative analyses to support much of the data discussed and/or shown. Additionally, the authors introduce three novel tools/resources in this paper: the morphometric quantification method and the BEST1-SIRT6-ERT2 and BEST1-NFKB1m-ERT2 transgenic mouse lines; however, these are either explained with minimal methodological detail and/or lack experimental validation.

2. Major Comments:

Figure 1: the authors state the use of a morphometric method developed to quantify each distinct region in RPE flat-mounts. Beyond stating this, however, there is no detailed methodology describing the morphometric quantification herein. The morphometric methodology is referenced in an already published paper by the same authors (Yang et al. PMID: 35052607) as "manuscript submitted", but not described in detail. The authors should provide detailed information about how this methodology was developed and utilized for this study, addressing questions such as: Is quantification blinded? What are the criteria for including/excluding cells at the very edges of each region? Specifically, what is the morphological criteria for "normal" RPE versus "EMT/transitional" RPE? What are the criteria for excluding RPE flat-mounts/areas of flat-mounts from analysis?

Figures 2 & 3: to support the claims made about regional RPE-specific (periphery) and, especially, EMT-specific (transitional zone) gene expression patterns, the authors should quantify the number of cells expressing SOX9, OTX2, SNAI1, ZEB1, and other RPE- and EMT-specific proteins in the peripheral, transitional, and central regions using their morphometric method. This is important as there does appear to be centrally expressed SOX9 at day 3 (Figure 2A), which contradicts the conclusion made in the text (p5). Similarly, SNAI1 is visibly expressed in all regions at day 3 (Figure 3A), and the authors should further define and quantify "weakly" when referencing SNAI1 expression in the periphery (p5).

Figures 2 & 3: the conclusions made about the Western blot results in both Figures 2 and 3 should be supported with band quantification, especially as the SOX9 band appears visibly larger at day 7 than at days 1 or 5 (Figure 2F), which would not support the mRNA expression results, as the authors state it does (p5). Additionally, the SNAI1 day 0 lane appears smeared/obstructed (Figure 3F) making it difficult to derive any conclusions about the expression level at this time point and this should be replaced with a better blot image.

Figure 4: the authors show that ZEB1 is expressed in the transitional zone (and centrally as well, it appears) on day 5 (Figure 3B). ZEB1 actually appears to be largely restricted to the peripheral RPE and not expressed in the transitional zone at day 3 (Figure 3B), so it is unclear how ZEB1 can be used as a marker for the transitional zone at this time point, as shown in Figure 4, and this should be clarified. The authors should also quantify regional TUNEL expression as a way to support claims of "sporadic" and "intense/extensive" expression in different zones.

Figure 5: regional zooms of ex vivo RPE flat-mounts should be included in Figure 5 (C - F) to support claims about morphological preservation of the RPE in the periphery.

Figure 6: SIRT6 and G3BP1 appear to be expressed in the choroid as well as in the RPE (Figure 6C). In their methods, the authors indicate that protein lysates for Western blots were derived from eyecups containing the RPE, choroid, and sclera. Given this, how do the authors know that SIRT6 and G3BP1 proteins are not degraded in the RPE but preserved in the choroid?

Figure 7: the authors should provide more careful validation of the BEST1-SIRT6-ERT2 transgenic line. Co-localization of increased inducible SIRT6 and loss of H3K9Ac with endogenous BEST1 in mouse RPE should be shown regionally to validate claims about expression/functional specificity.

Similar to the above point, careful validation of expression/functional specificity should be shown for the BEST1-NFKB1m-ERT2 transgenic line as well.

Figure 7: Western blot bands should be quantified to back up claims about SIRT6 expression differences between whole RPE and central RPE lysates as these are not visibly obvious.

Figure 8: the authors should include representative flat-mount and regional zoom images of the tissues being quantified in Figure 8A,B.

3. Minor Comments:

1. As done for RT-qPCR/cDNA quantification and parts of Figure 8A, the authors should provide biological replicate information for all experiments and indicate if the images shown in the figures are representative of the sample group(s) analyzed.
2. Adding dotted lines to images showing RPE region borders (e.g., Figure 2A,B; Figure 3A,B) would help readers delineate regional expression patterns.
3. Page 7, last sentence of first full paragraph - is nuclear SIRT6 lost as doses of NaIO₃ increase? Please clarify.
4. The authors should provide some background and/or a summary of NaIO₃ dosing as it relates to the extent of RPE damage and to the time points used for this study. In general, the time points used for this study should be more clearly justified and discussed in the results sections.
5. The authors should show all datapoints (biological replicates) for all RT-qPCR expression data.
6. Is an anatomical landmark (or other methodology) used to designate tissue in the "central" and "periphery" regions for the images shown in Figures 5 - 7?
7. More details about the imaging acquisition parameters (e.g., "short" exposure times and "low" exposure settings) should be included for the SIRT6 observations in Figures 5 - 7.

Reviewer #3 (Comments to the Authors (Required)):

Manuscript: "SIRT6 overexpression in the nucleus protects mouse retinal pigment epithelium from oxidative stress" by Yang et al.

General Points:

This is an interesting manuscript analyzing the mechanisms involved in RPE damage induced by exposure to the sodium iodate (NaIO₃) agent. The authors reported that elongated cells in the transitional zone exhibited molecular characteristics of epithelial-to-mesenchymal transition. Also, the authors determined that central RPE was more susceptible to stresses than peripheral RPE. Finally, the authors showed that SIRT6 overexpression in the nucleus protected RPE from NaIO₃ and preserved catalase expression in transgenic mice. The authors conclude that the presented data demonstrated topological differences in mouse RPE and suggested SIRT6 as a potential therapeutic target for oxidative stress-related RPE damage. The potential significance of these findings is high as NaIO₃ is the most used mouse model of RPE and photoreceptor degeneration. However, there are a few issues that need to be addressed by this study to improve the impact of the findings.

Specific Points:

- The text should be revised and edited for clarity.
- In Figures 2 and 3, the authors show low high magnification of the RPE flatmounts. It would be helpful to see higher magnification images.
- On page 7, the authors wrote: "The results provide clear experimental evidence supporting that photoreceptor death is mostly secondary to RPE death or dysfunction in NaIO₃-injected animals." The primary target cell of NaIO₃ is a highly debated topic. This is an over interpretation of the data presented; time-dependent analysis should be performed to determine the primary target of NaIO₃ or the text should be modified.
- Additional information about the SIRT6 and NFKB1m should be provided as RPE-transgenic mice expression are frequently patchy.
- Throughout the text, the authors state that SIRT6 has potential therapeutic application. However, the data presented data showed that SIRT6 did not reverse RPE damage. The text should be edited.

Response to Reviewers' Comments

First, we would like to express our enormous appreciation to the Reviewers for reading carefully and commenting on our manuscript with critical and constructive suggestions. Below we have tried to address such comments as much as we could. It has been long time since we received the comments. The major reason why it took so long was that we needed to breed and expand our transgenic mouse lines from the bare minimum during the COVID-19 pandemic. As described below in our response, we made a major change in the revised manuscript, i.e., we entirely removed the NFKB1m transgenic mice from the manuscript for multiple reasons, which we explain in detail below.

NOTE: The revision was substantial and therefore we made “**Summary of major changes**” at the end of this document. In addition, we submit the revised manuscript in Word with Track Changes (changes are marked in **red**) for your reference to see more easily where/what/how we changed.

Reviewer #1

1. Using a sodium iodate (NaIO₃)-mediated method to induce oxidative stress-related RPE and retinal damage in mice, Yang, Chung et al. investigated the expression patterns of RPE and EMT factors at different time points post-NaIO₃ injection. The authors also examined regional changes in cellular SIRT6 localization and SIRT6 co-localization with G3BP1 post-NaIO₃ injection. In addition, transgenic mice were used to determine the protective effects of nuclear SIRT6 over-expression and impaired nuclear NF-KB signaling on RPE after NaIO₃-mediated damage. While oxidative stress-induced RPE damage is an active and interesting area of research, this study is lacking quantitative analyses to support much of the data discussed and/or shown. Additionally, the authors introduce three novel tools/resources in this paper: the morphometric quantification method and the BEST1-SIRT6-ERT2 and BEST1-NFKB1m-ERT2 transgenic mouse lines; however, these are either explained with minimal methodological detail and/or lack experimental validation.

<Response>

Thank you so much for the constructive criticisms, comments, and suggestions on our manuscript. To respond to the general comments above, we made the following changes.

1) We added quantification of immunofluorescence and Western blotting to all relevant figures and the text as described below. 2) We described the method for quantifying RPE damage in more detail in the Results section. 3) We added more data of characterization of the *BEST1-SIRT6-ER^{T2}* transgenic mice to the text and figures as described in the response below. However, as mentioned above, we also made a major change, i.e., removed the *BEST1-NFKB1m-ER^{T2}* transgenic mice altogether, of which reasons are described in detail later. We truly hope that the Reviewer find these changes satisfactory and improving the manuscript.

2. Major Comments:

Figure 1: the authors state the use of a morphometric method developed to quantify each

distinct region in RPE flat-mounts. Beyond stating this, however, there is no detailed methodology describing the morphometric quantification herein. The morphometric methodology is referenced in an already published paper by the same authors (Yang et al. PMID: 35052607) as "manuscript submitted", but not described in detail. The authors should provide detailed information about how this methodology was developed and utilized for this study, addressing questions such as: Is quantification blinded? What are the criteria for including/excluding cells at the very edges of each region? Specifically, what is the morphological criteria for "normal" RPE versus "EMT/transitional" RPE? What are the criteria for excluding RPE flat-mounts/areas of flat-mounts from analysis?

<Response>

We thank the Reviewer for pointing this out. We must admit that our description was not enough. Therefore, we added more detailed description of the method for quantifying RPE damage in the Results section (page 5). We used "aspect ratio (AR)" as the criteria for separating normal-appearing RPE in the periphery and elongated cells in the transitional zone. AR is the ratio of the cell's major axis to the minor axis and used as one of the cell's morphological characteristics in other studies as well. RPE cells with compact cobblestone appearance have AR closer to 1, and elongated cells have AR above 1, particularly higher than 1.5. Therefore, we arbitrarily separated the two regions using AR = 1.5 as a cut-off. However, this separation was done by manual inspection, not by a computer-based format, and therefore it was approximate. In terms of the other paper the Reviewer mentioned (Yang et al., 2021), we were working on the two manuscripts in parallel, this study and the other. In reality, this study started earlier, and the other was developed later but unexpectedly published quickly (Yang et al., 2021). Thus, the timing of the two papers became in reverse order, which makes the situation awkward and confusing. Since the other paper was already published, we cited it more in this revised manuscript.

Figures 2 & 3: to support the claims made about regional RPE-specific (periphery) and, especially, EMT-specific (transitional zone) gene expression patterns, the authors should quantify the number of cells expressing SOX9, OTX2, SNAI1, ZEB1, and other RPE- and EMT-specific proteins in the peripheral, transitional, and central regions using their morphometric method. This is important as there does appear to be centrally expressed SOX9 at day 3 (Figure 2A), which contradicts the conclusion made in the text (p5). Similarly, SNAI1 is visibly expressed in all regions at day 3 (Figure 3A), and the authors should further define and quantify "weakly" when referencing SNAI1 expression in the periphery (p5).

<Response>

We thank the Reviewer for this important suggestion and totally agree with him/her. To respond to this comment as well as the Reviewer #3's request for higher magnification images, we first went through our collection of images and replaced all images of SOX9, OTX2, SNAI1, and ZEB1 with different zoomed images that we believe are clearer. Then, we quantified the number of positive cells (SOX9 and OTX2) and the signal intensity (SNAI1 and ZEB1) in the three regions on day 3, which are included in new Fig. 2c and Fig. 3c, respectively. The methods of these quantifications are described in Results and Figure Legends (marked in red by Track Changes). We would like to note that the quantification in the periphery was performed in the mid-far periphery in the original images, not near the transitional zone in the images shown in Figures 2 and 3. We chose these RPE- and EMT-markers because they are transcription factors (nuclear proteins), and therefore it is easier

to detect and count positive cells. Other commonly used RPE markers such as RPE65 and RLBP1 are cytoplasmic proteins and therefore diffusely stained, which makes more difficult to count/quantify.

As for SOX9 expression on day 3, fuzzy green staining in the center does not show a typical pattern of SOX9 nuclear staining as seen in the periphery (see the original images with zoomed areas A, B, and C below). We regret that our original images were misleading.

However, we did not regard this diffuse staining as functional SOX9 expression in the nucleus; rather, we interpreted it as non-specific staining because cells were severely damaged or lost and therefore unrecognizable in the center. Since RPE cell

damage and death happen in the center with cell debris scattering, we often observe diffuse high background in this region with many different antibodies.

With regard to SNAI1, its expression indeed increased in all regions on day 3 compared with the expression on day 0 when it was barely detectable. We think it important that SNAI1 expression increased even in normal-appearing RPE in the periphery, indicating that RPE cells in the periphery already show molecular changes suggestive of EMT. We quantified the signal intensity of SNAI1 immunofluorescence in randomly selected 20 DAPI stained nuclei in the three regions and found that SNAI1 levels were lower in the periphery compared with those in the transitional zone (Fig. 3c). We modified the description “weakly” in the Results section accordingly.

Figures 2 & 3: the conclusions made about the Western blot results in both Figures 2 and 3 should be supported with band quantification, especially as the SOX9 band appears visibly larger at day 7 than at days 1 or 5 (Figure 2F), which would not support the mRNA expression results, as the authors state it does (p5). Additionally, the SNAI1 day 0 lane appears smeared/obstructed (Figure 3F) making it difficult to derive any conclusions about the expression level at this time point and this should be replaced with a better blot image.

<Response>

We agree with the Reviewer that our Western blot results needed quantification and improvement. Since we originally performed Western blot analyses (days 0, 1, 5, 7) twice, we needed additional biological replicates for quantification. Therefore, we took this opportunity and performed completely new experiments for 3 sets of Western blots with an additional time point (days 0, 1, 3, 5, 7). We analyzed three biological replicates in 3 separate gels (Gels 1, 2, and 3), and the signal intensity of each band was quantified using ImageJ. Gel 1 and the quantification results for RPE markers and EMT factors are shown in new Fig. 2f and Fig. 3g, with Gels 2 and 3 being included in Fig. S1a and b, respectively.

As for Western blot analyses with EMT-related factors, we could not detect significant differences in protein levels using the whole RPE lysates, except the modest increase of α -SMA on day 7 (Fig. 3g). It was particularly disappointing and conflicting that while vimentin increased at the mRNA level on day 7, its protein level was not changed. We

speculated that some molecular changes might have already occurred even in the periphery, which was detected by sensitive RT-qPCR but not by Western blotting, and that analyses of whole RPE lysates might not be the best approach due to regional differences of RPE damage. Therefore, we performed immunofluorescence (IF) for vimentin (VIM) with RPE flat-mounts on day 7 after NaIO₃ injection to detect localized differences in VIM expression. Indeed, VIM protein significantly increased in the transitional zone and center, and these results are newly added as Fig. 3h (IF), Fig. 3i (quantification results), and Fig. S2 (strategy of quantification).

Figure 4: the authors show that ZEB1 is expressed in the transitional zone (and centrally as well, it appears) on day 5 (Figure 3B). ZEB1 actually appears to be largely restricted to the peripheral RPE and not expressed in the transitional zone at day 3 (Figure 3B), so it is unclear how ZEB1 can be used as a marker for the transitional zone at this time point, as shown in Figure 4, and this should be clarified. The authors should also quantify regional TUNEL expression as a way to support claims of "sporadic" and "intense/extensive" expression in different zones.

<Response>

We thank the Reviewer for reading the manuscript very carefully and raising an important issue. We admit and are very sorry for our poor selection of the ZEB1 images in the original

Fig. 3b. In this specific experiment, we were so impressed with the intensity of ZEB1 signals in the transitional zone on day 5 that we wanted to show that particular image. However, since we had multiple other images showing increased expression of ZEB1 in the transitional zone on day 3, the original ZEB1 images in Fig. 3b were not representative but rather atypical. Therefore, we should have used more typical images (see Samples 1–3 from 3 independent experiments). Accordingly, we replaced the original images with the images of Sample 1.

As for the need of quantifying TUNEL signals, we totally agree with the Reviewer. Due to the difficulties in counting TUNEL-positive cells, we instead quantified the signal intensity

of TUNEL staining using ImageJ (new Fig. 4e, f). As described in Results and Figure Legends, we set the squared area for quantification and slid it horizontally on the retinal TUNEL image. The total signal intensity at different locations shows that TUNEL signals are positive but weaker in the transitional zone that corresponds to increased ZEB1 expression (locations 6–8 in Fig. 4f). Based on these results, we replaced the description "sporadic" with "moderate" and "weaker" as to photoreceptor cell death and TUNEL signals, respectively, in the Results section (page 8).

Figure 5: regional zooms of ex vivo RPE flat-mounts should be included in Figure 5 (C - F) to support claims about morphological preservation of the RPE in the periphery.

<Response>

Thank you for this good suggestion. We added zoomed images from the center (area A) and periphery (area B) of each ex vivo RPE flat-mount (Fig. 5c-f).

Figure 6: SIRT6 and G3BP1 appear to be expressed in the choroid as well as in the RPE (Figure 6C). In their methods, the authors indicate that protein lysates for Western blots were derived from eyecups containing the RPE, choroid, and sclera. Given this, how do the authors know that SIRT6 and G3BP1 proteins are not degraded in the RPE but preserved in the choroid?

<Response>

We would like to clarify our method for preparing protein lysates for Western blots. We used the method reported by Wei, et al. to prepare RPE protein lysates from eyecups consisting of the RPE, choroid, and sclera by incubating them in RIPA lysis buffer on ice for 45 min with occasional gentle tapping to release RPE proteins without homogenization or sonication (Wei, et al., 2016). This gentle method yields RPE protein lysates with only minor contamination of choroidal proteins because while RPE cells are located on the surface of the eyecups, the choroid is hidden between the Bruch's membrane and sclera, both of which are made of strong collagenous fibers. Thus, the results of Western blots with our RPE protein lysates mainly reflect RPE proteins.

Figure 7: the authors should provide more careful validation of the BEST1-SIRT6-ERT2 transgenic line. Co-localization of increased inducible SIRT6 and loss of H3K9Ac with endogenous BEST1 in mouse RPE should be shown regionally to validate claims about expression/functional specificity.

Similar to the above point, careful validation of expression/functional specificity should be shown for the BEST1-NFKB1m-ERT2 transgenic line as well.

<Response>

We appreciate the Reviewer for raising this important point to improve our manuscript. We would like to respond as follows.

First, we would like to clarify that the *BEST1* -585 to +38 bp promoter (*BEST* -585/+38) is a human promoter. When we characterized this promoter, we noticed that the expression mode was different between human *BEST1* and mouse *Best1*. While *BEST1* was highly expressed in the RPE of human donor eyes, the expression of *Best1* in the RPE of mouse eyes was much lower, which was also confirmed by the number of clones in RPE cDNA libraries from each species (unpublished data). By comparing the genomic sequences, we found that human and mouse genes utilize different genomic regions as their promoters, i.e., the mouse *Best1* promoter is located at about 5 kb upstream of the region corresponding to the human *BEST1* promoter, which we think could explain, at least in part, the differences in the expression mode between two species.

Second, based on the findings described above, we assumed that the expression of endogenous mouse BEST1 would not be informative for inducible SIRT6 expression in our transgenic mice. Therefore, we instead added an old but unpublished result of whole RPE

flat-mount with β -galactosidase (*lacZ*) staining from the *BEST1* –585/+38 promoter–*lacZ* transgenic mice to show the spatial characteristics of this promoter in mouse RPE (new Fig. 7a). Then, we added RPE flat-mounts with SIRT6 immunofluorescence from wild-type mice and SIRT6 transgenic mice (S4 line) injected with either vehicle or tamoxifen (Tmx) to show the spatial pattern of SIRT6 induction (new Fig. 7b). The distribution of induced SIRT6 was similar to the pattern of *lacZ* expression driven by the same promoter that is more active in the center than in the periphery. This center dominant induction of SIRT6 seemed to give the benefits for protecting central RPE where NaIO_3 causes more severe damage. Additionally, we also observed the leaky expression of SIRT6 even with vehicle in our transgenic mice.

Third, we performed double immunofluorescence for SIRT6 and H3K9Ac on RPE flat-mounts. Previously we analyzed SIRT6 and H3K9Ac separately because the antibodies for these proteins in our possession were both rabbit antibodies. With a newly purchased mouse monoclonal antibody for H3K9Ac, we could show regional differences in the SIRT6 induction and its effect on H3K9Ac levels (new Fig. 7d, e). In most of the periphery, SIRT6 was not induced with Tmx, and strong staining of H3K9Ac was observed in general. In peripheral RPE cells with a sporadic SIRT6 induction, H3K9Ac staining was generally less intense. In the center, SIRT6 was induced with Tmx in the RPE nuclei in large areas, and H3K9Ac staining was weak in such nuclei in general. In contrast, in central RPE cells with a poor SIRT6 induction, H3K9Ac levels tended to be higher in such cells. Then, because we provided these new double immunofluorescence images of RPE flat-mounts, and because the previous retinal sections (old Fig. 7a) did not have the regional information about SIRT6 and H3K9Ac levels, we deleted the images of retinal sections from the revised manuscript.

Fourth, as for the NFKB1m transgenic mice, we admit that we did not show enough validation data for these mice. We had tried to obtain the expression-function relationship in the similar manner to that for the SIRT6 transgenic mice as described above. However, because the basal NF- κ B activity is very low in the unstimulated state, and NF- κ B signaling is dynamic, it was difficult to show the expression-function specificity at the cellular level regardless of basal or stimulated states. That was the reason why we previously used RT-qPCR with whole RPE RNAs to detect suppression of the upregulation of NF- κ B targets after NaIO_3 injection by NFKB1m induction. Thus, these RT-qPCR analyses aimed at both functional validation of the NFKB1m transgenic mice and actual experiments for testing the effects of blocking NF- κ B on oxidative stress-caused RPE damage. We have tried again double immunofluorescence for NFKB1 and ICAM1, one of NF- κ B targets. Regrettably, however, we could not obtain clear results enough to include in the revised manuscript. We discussed and decided to **remove the NFKB1m transgenic mice** from this manuscript. Additional reasons are: 1) the focus of this study is on SIRT6, 2) NF- κ B plays only a supplementary negative role in understanding the mechanisms of SIRT6's function against oxidative stress in this study, 3) even if we eliminate the NFKB1m mice, we can still refer to the results of IKK β inhibitor (upstream inhibitor of NF- κ B signaling) in our recent paper (Yang et al., 2021), 4) more data and analyses related to SIRT6 were added, making the already long manuscript longer, and 5) increasing the focus on SIRT6 would make this manuscript more concise and clearer. We hope that the Reviewers and Editor understand and take this major change positively.

Figure 7: Western blot bands should be quantified to back up claims about SIRT6 expression differences between whole RPE and central RPE lysates as these are not visibly obvious.

<Response>

We totally agree with the Reviewer that our Western blot results needed quantification. Therefore, we collected additional samples, analyzed them by another Western blot (Gel 2 in Fig. S1d), and quantified Western bands using ImageJ as we did with other Western blot bands. We observed a modest but significant difference in SIRT6 levels between whole RPE and central RPE. We speculate that endogenous mouse SIRT6 likely masked induction of human SIRT6 to some extent, resulting in the modest difference.

Figure 8: the authors should include representative flat-mount and regional zoom images of the tissues being quantified in Figure 8A,B.

<Response>

Yes, we included representative images of RPE flat-mounts with ZO-1 immunofluorescence and zoomed images of the three regions from these RPE flat-mounts (new Fig. 8a).

3. Minor Comments:

1. As done for RT-qPCR/cDNA quantification and parts of Figure 8A, the authors should provide biological replicate information for all experiments and indicate if the images shown in the figures are representative of the sample group(s) analyzed.

<Response>

Yes, we added the information about the number of biological replicates for all experiments as (n = XX for each group) in Figure Legends. We also included “Representative images are shown” wherever appropriate in Figure Legends.

2. Adding dotted lines to images showing RPE region borders (e.g., Figure 2A,B; Figure 3A,B) would help readers delineate regional expression patterns.

<Response>

Thank you for this great suggestion to make our results clearer. Yes, we added dotted lines for the border of the three regions in Figures 2a, 2b, 3a, 3b, 3h, and S2.

3. Page 7, last sentence of first full paragraph - is nuclear SIRT6 lost as doses of NaIO₃ increase? Please clarify.

<Response>

Yes, that is correct in our experimental conditions with 8–12-week-old male C57BL/6J mice. In the periphery, SIRT6 was dominantly localized in the nucleus with NaIO₃ up to 10 mg/kg body weight (BW), but nuclear SIRT6 decreased with NaIO₃ at 15 mg/kg BW. In the center, nuclear SIRT6 began to decrease already with NaIO₃ at 10 mg/kg BW and further decreased at 15 mg/kg BW (Fig. 5a). We quantified the intensity of SIRT6 signals in the DAPI-stained nuclei (20 nuclei for each condition) using ImageJ and added the results to Fig. 5a. There was no statistically significant difference in SIRT6 signals between 15 and 20 mg/kg BW of NaIO₃ in either the periphery or the center, suggesting that NaIO₃ at 15 mg/kg BW may exert a near maximum effect regarding the loss of nuclear SIRT6.

4. The authors should provide some background and/or a summary of NaIO₃ dosing as it

relates to the extent of RPE damage and to the time points used for this study. In general, the time points used for this study should be more clearly justified and discussed in the results sections.

<Response>

Thank you for pointing out these important issues about the dose of NaIO₃ and the time points of analyses. We should have discussed these issues more clearly in the manuscript. But now we have a chance to correct our shortcomings and would like to respond as below.

As for NaIO₃ dosing, we tested the effects of different doses of NaIO₃ (10–20 mg/kg BW) on RPE morphological damage in parallel with this study and reported the results in a separate publication (Yang et al., 2021). With male C57BL/6J mice, we found a narrow window (11–18 mg/kg BW) of NaIO₃ dose-effect correlation as follows:

- 10 mg/kg BW – no clear morphological RPE damage
- 11 mg/kg BW – minimum dose (threshold) causing a morphological RPE damage
- 15 mg/kg BW – severe RPE damage in roughly a half of the RPE
- 18 mg/kg BW – severe RPE damage in the nearly entire RPE
- 20 mg/kg BW – severe RPE damage in the nearly entire RPE

Based on these results, we used NaIO₃ at 15 mg/kg BW in most of the experiments in this study. However, we also used different doses of NaIO₃ in the specific experiments as summarized below. We added these background and summary in the Results and Methods sections (page 5 and page 18, respectively).

NaIO ₃ dose (mg/kg BW)	Experiments/Analyses
15	RPE damage (RPE flat-mounts), retinal degeneration (TUNEL assays), mRNA expression (RT-qPCR), protein expression (Western blots)
0, 5, 10, 15, and 20	dose-dependence of nuclear SIRT6 loss
20*	SIRT6 translocation from the nucleus to the cytoplasm
60**	analyses of the effects of SIRT6 induction on gene expression

* We used this dose to give a stronger oxidative stress than 15 mg/kg BW to make certain that SIRT6 translocation could occur more clearly.

** We used this high dose to upregulate NF-κB target genes to the larger degrees so that repression of the upregulation could be more easily detected.

Regarding the time points, we analyzed the extent of RPE damage on day 7 after NaIO₃ injection throughout this study. We would like to clarify and justify why/how we chose this time point. First, we conducted preliminary studies using RPE flat-mounts with ZO-1 immunofluorescence at earlier time points (days 0, 1, 2, 3, 4, 5, and 7) and later time points (2 weeks and 1 and 3 months) after NaIO₃ injection. The RPE was fragile and easily broken when making RPE flat-mounts before day 3. The transitional zone was not present on days 1 and 2 but formed by day 3. The RPE structure became more solid and stabilized by days 5–7, when we could make better RPE flat-mounts more easily. These findings at earlier time points were partially described in our previous publication (Yang et al., 2018). We followed RPE damage after day 7, and the proportion of the three regions did not significantly change at least up to 3 months. Therefore, the earliest time point to analyze RPE flat-mounts with ease and consistency was day 7, which we chose as the universal time point for this study. We added this background for the choice of day 7 for analyzing RPE damage in the Results section (page 4–5). In addition, the choices of time points for analyzing gene expression by RT-qPCR, protein expression by immunofluorescence and Western blotting, and SIRT6 translocation by immunofluorescence were discussed at each experiment whenever appropriate or desirable in the Results section.

5. The authors should show all datapoints (biological replicates) for all RT-qPCR expression data.

<Response>

Thank you for pointing this out. Yes, we included all datapoints in the graphs of all RT-qPCR results. Some of the results were from analyses more than 8 years ago, and we could not find the original individual data. In such cases, we re-analyzed and obtained all individual datapoints.

6. Is an anatomical landmark (or other methodology) used to designate tissue in the "central" and "periphery" regions for the images shown in Figures 5 - 7?

<Response>

We appreciate the Reviewer's careful reading of our manuscript. We defined the optic nerve head as the center, which is the center of whole RPE flat-mounts. Then, the whole RPE flat-mount is divided into 3 concentric zones from the periphery to the center, with one-third of the length of a radial line for each zone, periphery, middle, and center (a, b, and c, respectively in the image below). We added this explanation in the Results section (page 8).

7. More details about the imaging acquisition parameters (e.g., "short" exposure times and "low" exposure settings) should be included for the SIRT6 observations in Figures 5 - 7.

<Response>

Thank you for the suggestion. Yes, we included the microscopic parameters for acquiring SIRT6 images (Alexa 546) in the Figure Legends (marked in red by Track Changes).

Reviewer #3

Manuscript: "SIRT6 overexpression in the nucleus protects mouse retinal pigment epithelium from oxidative stress" by Yang et al.

General Points:

This is an interesting manuscript analyzing the mechanisms involved in RPE damage induced by exposure to the sodium iodate (NaIO₃) agent. The authors reported that elongated cells in the transitional zone exhibited molecular characteristics of epithelial-to-mesenchymal transition. Also, the authors determined that central RPE was more

susceptible to stresses than peripheral RPE. Finally, the authors showed that SIRT6 overexpression in the nucleus protected RPE from NaIO₃ and preserved catalase expression in transgenic mice. The authors conclude that the presented data demonstrated topological differences in mouse RPE and suggested SIRT6 as a potential therapeutic target for oxidative stress-related RPE damage. The potential significance of these findings is high as NaIO₃ is the most used mouse model of RPE and photoreceptor degeneration. However, there are a few issues that need to be addressed by this study to improve the impact of the findings.

Specific Points:

-- The text should be revised and edited for clarity.

<Response>

Thank you for the comment. First, we believe that the removal of the NFKB1m transgenic mice has made this manuscript more focused and clearer with a better flow. In addition, we went through the entire manuscript and modified the text to make it as clear as possible. Furthermore, in response to both Reviewers' comments, we added many descriptions to explain the rationales and processes in more detail to reduce the vagueness. We believe these changes have made the manuscript clearer.

-- In Figures 2 and 3, the authors show low high magnification of the RPE flatmounts. It would be helpful to see higher magnification images.

<Response>

We very much appreciate this suggestion. We went through our collection of image files and replaced all images of RPE flat-mounts in the previous Figures 2 and 3 with different sets of images. At the same time, we also added higher magnification images of SOX9, OTX2, SNAI1, and ZEB1 to each RPE flat-mount image on Day 3. We believe that these changes made our figures clearer and easier to interpret the results. Please also see our response to the Reviewer #1 about Figures 2 & 3.

-- On page 7, the authors wrote: "The results provide clear experimental evidence supporting that photoreceptor death is mostly secondary to RPE death or dysfunction in NaIO₃-injected animals." The primary target cell of NaIO₃ is a highly debated topic. This is an over interpretation of the data presented; time-dependent analysis should be performed to determine the primary target of NaIO₃ or the text should be modified.

<Response>

Thank you for careful reading and pointing out this mistake. We admit our over-interpretation of the results, and therefore we have modified the text as below (page 8). "These results seem to support the hypothesis that photoreceptor death is mostly secondary to RPE death or dysfunction in NaIO₃-injected animals."

-- Additional information about the SIRT6 and NFKB1m should be provided as RPE-transgenic mice expression are frequently patchy.

<Response>

We greatly appreciate the Reviewer for raising the important issue. Both Reviewers #1 and #3 raised this issue, and we recognize that our data were insufficient for characterization of the two transgenic mouse models. We described our response to this issue in detail in the “Reviewer #1” section above. Therefore, we briefly summarize our responses here.

1) We used the human *BEST1* –585 to +38 bp promoter (*BEST* –585/+38) for our transgenic mice. We included an old but unpublished result of whole RPE flat-mount with β -galactosidase (*lacZ*) staining from the *BEST1* –585/+38 promoter–*lacZ* transgenic mice, which showed patchy spatial characteristics of this promoter in mouse RPE (new Fig. 7a).

2) We added RPE flat-mounts with SIRT6 immunofluorescence from wild-type mice and SIRT6 transgenic mice with vehicle or tamoxifen (Tmx). The spatial patterns of SIRT6 induction with Tmx were similar to those of *lacZ* staining described above (new Fig. 7b).

3) We added images of double immunofluorescence for SIRT6 and H3K9Ac of RPE flat-mounts. The images showed regional differences (periphery vs. center) in the induction of SIRT6 and its effect on H3K9Ac levels (new Fig. 7d, e). In the same cells, the levels of SIRT6 and H3K9Ac tended to be inversely correlated in general.

4) We also collected more samples of RPE lysates for Western blots, which showed that SIRT6 was induced at higher levels in the central region (new Fig. 7c).

5) We deleted the previous images of retinal sections (old Fig. 7a) that lacked the regional information about SIRT6 and H3K9Ac levels.

6) Major change: we **removed the NFKB1m transgenic mice** from this manuscript for multiple reasons. This decision was made because i) we could not show clear expression-function relationships at the cellular level, ii) SIRT6 was the focus of this study, and NF- κ B played only a minor and negative role, iii) we added more data and analyses related to SIRT6, which made the manuscript quite long, and iv) we wanted to make this manuscript more concise and clearer by increasing the focus on SIRT6.

Again, we hope that the Reviewers understand and take these changes positively.

-- Throughout the text, the authors state that SIRT6 has potential therapeutic application. However, the data presented data showed that SIRT6 did not reverse RPE damage. The text should be edited.

<Response>

Thank you very much for pointing this out. We agree with the Reviewer. SIRT6 seems to protect RPE at the early stages of oxidative damage by preserving catalase expression and associating with stress granule formation. Thus, SIRT6 could be helpful for prevention, but not treatment, of oxidative damage. Therefore, we modified the text and eliminated our original statement throughout the manuscript.

Summary of major changes

Changes in Figures

Figure 1

- No change in the figure

Figure 2

- Replaced immunofluorescence images for SOX9 and OTX2
- Added higher magnification images for SOX9 and OTX2
- Added dotted lines for the border of the three regions
- Quantified the number of cells expressing SOX9 and OTX2 in the images
- Showed all data points (biological replicates) for RT-qPCR expression data
- Performed new Western blotting on days 0, 1, 3, 5, 7 after NaIO₃ injection
- Quantified Western blot bands

Figure 3

- Replaced immunofluorescence images for SNAI1 and ZEB1
- Added higher magnification images for SNAI1 and ZEB1
- Added dotted lines for the border of the three regions
- Quantified the signal intensity of SNAI1 and ZEB1 in the images
- Showed all data points (biological replicates) for RT-qPCR expression data
- Performed new Western blotting on days 0, 1, 3, 5, 7 after NaIO₃ injection
- Quantified Western blot bands
- Added immunofluorescence images of VIM on day 7 and quantified signals

Figure 4

- Quantified the signal intensity of TUNEL staining

Figure 5

- Quantified the signal intensity of SIRT6 with different doses of NaIO₃
- Quantified the signal intensity of SIRT6 in fresh and *ex vivo* RPE
- Added zoomed images of *ex vivo* RPE with oxidative stress reagents

Figure 6

- Performed more Western blotting at 0, 0.5, 1, 2, 6, and 12 h after NaIO₃ injection
- Quantified Western blot bands for RPE65, SIRT6, and G3BP1

Figure 7

- Added the construct of SIRT6 transgenic mice
- Added a lacZ-stained image from the *BEST1* -585/+38 promoter-*lacZ* mice
- Added new RPE flat-mounts images with induced SIRT6 (vehicle vs. Tmx)
- Deleted the images of retinal sections due to the lack of regional information.
- Quantified Western blot bands with additional Western blotting for SIRT6 induction
- Added new RPE flat-mounts images with SIRT6 induction and H3K9Ac loss

Figure 8

- Removed all data related to the NFκB1m transgenic mice
- Added representative images of RPE flat-mounts of SIRT6 mice (vehicle vs. Tmx)

-- Showed all data points (biological replicates) for RT-qPCR expression data

Figure S1 (new) – supplements to Figures 2f, 3g, 6b, and 7c

- All Western blots for additional biological replicates
RPE markers (Fig. 2f) and EMT-related proteins (Fig. 3g)
SIRT6 and G3BP1 time-course (Fig. 6b)
SIRT6 induction by Tmx in SIRT6 transgenic mice (Fig. 7c)

Figure S2 (new) – supplements to Figure 3i

- Strategy to quantify the signal intensity of vimentin (VIM) immunofluorescence

Figure S3 (new) – supplements to Figure 6a

- Quantification of SIRT6 signal intensity after NaIO₃ injection – time course

Figure S4 (new) – supplements to Figure 7g

- Quantification of the signal intensity of SIRT6 and H3K9Ac after NaIO₃ injection

Figure S5 (old S1)

- Removed the plasmid construct

Figure S6 (old S2)

- Removed all data from the treatment scheme with the NFκB1m transgenic mice
-- Added representative images of RPE flat-mounts of SIRT6 mice (vehicle vs. Tmx)

Changes in Results

As summarized above, we made substantial changes in Figures. Accordingly, we revised a large portion of the text to explain these changes. Major changes are summarized below.

-- Because we completely removed NFκB1m transgenic mice from this manuscript, we revised to reflect this change at multiple sections.

-- Throughout the manuscript, we quantified immunofluorescence signals as much as possible beyond the Reviewers' comments and described such results in the text.

-- We also quantified all Western blot bands and described the results at appropriate locations in the text.

-- We explained the background for the doses of NaIO₃ and the time points of analyses and described the details of the morphometric quantification method (related to Figure 1).

-- We added new immunofluorescence images of vimentin (VIM) to address the conflicting results of RT-qPCR and Western blotting and discussed the results (related to Figure 3). This was not in the Reviewers' comments.

-- We added the information of human *BEST1* -585/+38 promoter and explained the spatial characteristics of this promoter that we used to generate our SIRT6 transgenic mice. This was also not in the Reviewers' comments.

-- We added new data/images for characterization of SIRT6 transgenic mice and described their results and interpretation in the text (related to Figure 7).

Changes in Discussion

We deleted some description related to NF- κ B. We also updated and modified Discussion by adding recent publications in the topics below.

-- Topological differences of RPE cells. A recent paper revealed five different RPE subpopulations locating in concentric circles in human RPE.

-- RPE EMT. A recent paper reported that impaired mitophagy and mitochondrial dysfunction triggered RPE EMT, leading to upregulation of SNAI1 and ZEB1.

-- SIRT6 and stress granules (SGs). SGs are formed by liquid-liquid phase separation (LLPS), and proteins undergoing LLPS contain intrinsically disordered regions (IDRs). SIRT6 is predicted to contain an IDR at the C-terminus, suggesting that SIRT6 could physically participate in LLPS to form SGs.

March 29, 2023

RE: Life Science Alliance Manuscript #LSA-2022-01448-TR

Dr. Noriko Esumi
Johns Hopkins University
Ophthalmology
400 North Broadway
Smith Building, Room 3041
Baltimore, Maryland 21231

Dear Dr. Esumi,

Thank you for submitting your revised manuscript entitled "SIRT6 overexpression in the nucleus protects mouse retinal pigment epithelium from oxidative stress". We would be happy to publish your paper in Life Science Alliance pending final revisions necessary to meet our formatting guidelines.

- please address the remaining Reviewer 3's concerns
- please add the Twitter handle of your host institute/organization as well as your own or/and one of the authors in our system
- please use the [10 author names, et al.] format in your references (i.e. limit the author names to the first 10)

Figure Check:

There are missing scale bars throughout. Please add scale bars to:

figure 1 all panels
figure 2A,B
figure 3 A,B,H
Figure 4 all panels
figure 5 all panels
Figure 6 A,C,D
Figure 7 B,D,E,F,G
FIGURE 8 A
Figure S2 all panels
figure S6A

A. FINAL FILES:

B. MANUSCRIPT ORGANIZATION AND FORMATTING:

Sincerely,

Reviewer #3 (Comments to the Authors (Required)):

Manuscript titled: "SIRT6 overexpression in the nucleus protects mouse retinal pigment epithelium from oxidative stress" by Yang et al.

This interesting manuscript analyzes the mechanisms involved in RPE damage induced by exposure to the sodium iodate (NaIO₃) agent. The NaIO₃ is the most used mouse model of RPE and photoreceptor degeneration. Thus, identifying potential intrinsic properties of RPE in response to this oxidizing agent is novel and very relevant to our understanding of it.

The authors extensively revised the original manuscript, and thus, the updated version improved significantly. However, there are a few issues that this reviewer believes should be addressed before the manuscript is accepted.

1- There is a bit of inconsistency in how the authors cite references, with texts describing similar observations citing different references. An example, in the introduction, the authors wrote: "Recently, several others and we also observed similar RPE morphologies induced by NaIO₃ in mice (Ma et al., 2020; Upadhyayet al., 2020; Wolk et al., 2020; Yang et al., 2021; Zhang et al., 2021)." While in the description of figure 1, the author wrote: "During these preliminary studies, we also observed that RPE damage by NaIO₃ was not evenly distributed as previously noted (Ma et al., 2020; Xia et al., 2011; Zhang et al., 2021). While RPE in the periphery was well preserved, RPE in the center (posterior) around the optic nerve head was severely damaged, either degenerated or completely lost, compared with control mice with vehicle (Fig. 1a, b)." The authors should revise the literature cited for consistency.

2- The author wrote in the description of the results of figure 1: "In this separate study, we tested the effects of different doses of NaIO₃ and found a narrow range of NaIO₃ dose-effect correlation regarding RPE morphological damage from 11 to 18 mg/kg body weight (BW) in male C57BL/6J mice (Yang et al., 2021). Because NaIO₃ at 15 mg/kg BW was within this narrow range and caused severe RPE damage in roughly half of the RPE, we assumed that this dose was more sensitive to modulating conditions and therefore used it in most of the experiments in the present study." This reviewer did not review the previous manuscript. Thus he/she will not criticize the manuscript. The understanding of this reviewer is that the

concentration of NaIO₃ injected in each mouse is achieved by injecting (from a 1 or 2% stock solution) a volume according to the mouse's weight. The injections are performed using syringes with 0.1ml graduation volume. Thus, this reviewer finds it very hard to inject NaIO₃ precisely at a concentration from 11 to 18 mg/Kg (excluding 15 mg/Kg). If the authors believe that these injections were precise, they should describe a detailed description of how the injections were performed. Otherwise, the reviewer would recommend the authors remove this information and discussion from the present manuscript.

3- Scale bars are missing from several figures. Figures should be edited.

Response to Reviewers' Comments

First, we would like to thank the Reviewers for positive comments on our first revised manuscript. Below we have tried to address the Reviewer 3's remaining concerns.

Reviewer #3

Manuscript titled: "SIRT6 overexpression in the nucleus protects mouse retinal pigment epithelium from oxidative stress" by Yang et al.

This interesting manuscript analyzes the mechanisms involved in RPE damage induced by exposure to the sodium iodate (NaIO₃) agent. The NaIO₃ is the most used mouse model of RPE and photoreceptor degeneration. Thus, identifying potential intrinsic properties of RPE in response to this oxidizing agent is novel and very relevant to our understanding of it. The authors extensively revised the original manuscript, and thus, the updated version improved significantly. However, there are a few issues that this reviewer believes should be addressed before the manuscript is accepted.

<Response>

We greatly appreciate the Reviewer's positive review of our revised manuscript. Please see our response to your remaining concerns.

1- There is a bit of inconsistency in how the authors cite references, with texts describing similar observations citing different references. An example, in the introduction, the authors wrote: "Recently, several others and we also observed similar RPE morphologies induced by NaIO₃ in mice (Ma et al., 2020; Upadhyayet al., 2020; Wolk et al., 2020; Yang et al., 2021; Zhang et al., 2021)." While in the description of figure 1, the author wrote: "During these preliminary studies, we also observed that RPE damage by NaIO₃ was not evenly distributed as previously noted (Ma et al., 2020; Xia et al., 2011; Zhang et al., 2021). While RPE in the periphery was well preserved, RPE in the center (posterior) around the optic nerve head was severely damaged, either degenerated or completely lost, compared with control mice with vehicle (Fig. 1a, b)." The authors should revise the literature cited for consistency.

<Response>

Thank you for the careful review and pointing this out. We corrected the inconsistency of references cited.

2- The author wrote in the description of the results of figure 1: "In this separate study, we tested the effects of different doses of NaIO₃ and found a narrow range of NaIO₃ dose-effect correlation regarding RPE morphological damage from 11 to 18 mg/kg body weight (BW) in male C57BL/6J mice (Yang et al., 2021). Because NaIO₃ at 15 mg/kg BW was within this narrow range and caused severe RPE damage in roughly half of the RPE, we assumed that this dose was more sensitive to modulating conditions and therefore used it in most of the experiments in the present study." This reviewer did not review the previous

manuscript. Thus he/she will not criticize the manuscript. The understanding of this reviewer is that the concentration of NaIO₃ injected in each mouse is achieved by injecting (from a 1 or 2% stock solution) a volume according to the mouse's weight. The injections are performed using syringes with 0.1ml graduation volume. Thus, this reviewer finds it very hard to inject NaIO₃ precisely at a concentration from 11 to 18 mg/Kg (excluding 15 mg/Kg). If the authors believe that these injections were precise, they should describe a detailed description of how the injections were performed. Otherwise, the reviewer would recommend the authors remove this information and discussion from the present manuscript.

<Response>

We certainly understand and agree with the Reviewer's concern. As suggested, we removed the previous results with NaIO₃ at 11 to 18 mg/kg BW and modified the rationale of our choice of NaIO₃ doses as below.

"In this separate study, we found that while NaIO₃ at 10 mg/kg body weight (BW) did not produce RPE morphological damage, NaIO₃ at 20 mg/kg BW caused severe RPE damage in the nearly entire RPE in male C57BL/6J mice (Yang et al., 2021). Because NaIO₃ at 15 mg/kg BW was between these two doses and caused severe RPE damage in roughly a half of the RPE, we assumed that this dose was more sensitive to modulating conditions and therefore used it in most of the experiments in the present study (page 5, lines 19–24)."

Just for the information for the Reviewer, we describe how we injected NaIO₃ here. We first made a stock solution (NaIO₃ 20 mg/mL PBS), which was a concentration of >10x of the final working solution for injection. From this stock solution, we made each working solution by diluting with PBS, e.g., 1.0 mg/mL (injection for 10 mg/kg BW), 1.1 mg/mL (for 11 mg/kg BW), 1.2 mg/mL (for 12 mg/kg BW), - - - 2.0 mg/mL (for 20 mg/kg BW), etc. If a mouse weighed 20 g, we injected 200 µL, which gave the intended dose. If a mouse weighed 23 g, we injected 230 µL, which gave the intended dose, and so on. We used Insulin syringes (max 0.5 mL) that had 10 µL graduation volume. Although we admit that our injection was not as precise as we wished, it was probably more accurate than the Reviewer thinks. In fact, we got a nice surprise that we could still obtain the reasonable dose-effect correlation with some variabilities using this injection procedure (Yang et al., 2021).

3- Scale bars are missing from several figures. Figures should be edited.

<Response>

Thank you for pointing this out. We added scale bars to all images in our figures.

April 6, 2023

RE: Life Science Alliance Manuscript #LSA-2022-01448-TRR

Dr. Noriko Esumi
Johns Hopkins University
Ophthalmology
400 North Broadway
Smith Building, Room 3041
Baltimore, Maryland 21231

Dear Dr. Esumi,

Thank you for submitting your Research Article entitled "SIRT6 overexpression in the nucleus protects mouse retinal pigment epithelium from oxidative stress". It is a pleasure to let you know that your manuscript is now accepted for publication in Life Science Alliance. Congratulations on this interesting work.

DISTRIBUTION OF MATERIALS:

Again, congratulations on a very nice paper. I hope you found the review process to be constructive and are pleased with how the manuscript was handled editorially. We look forward to future exciting submissions from your lab.

Sincerely,
